# AN ALGORITHM FOR OUT-OF-DISTRIBUTION ATTACK TO NEURAL NETWORK ENCODER

## ABSTRACT

Deep neural networks (DNNs), especially convolutional neural networks, have achieved superior performance on image classification tasks. However, such performance is only guaranteed if the input to a trained model is similar to the training samples, i.e., the input follows the probability distribution of the training set. Out-Of-Distribution (OOD) samples do not follow the distribution of training set, and therefore the predicted class labels on OOD samples become meaningless. Classification-based methods have been proposed for OOD detection; however, in this study we show that this type of method has no theoretical guarantee and is practically breakable by our OOD Attack algorithm because of dimensionality reduction in the DNN models. We also show that Glow likelihood-based OOD detection is breakable as well.

## 1 INTRODUCTION

Deep neural networks (DNNs), especially convolutional neural networks (CNNs), have become the method of choice for image classification. Under the i.i.d. (independent and identically distributed) assumption, a high-performance DNN model can correctly-classify an input sample as long as the sample is "generated" from the distribution of training data. If an input sample is not from this distribution, which is called Out-Of-Distribution (OOD), then the predicted class label from the model is meaningless. It would be great if the model has the ability to distinguish OOD samples from in-distribution samples. OOD detection is needed especially when applying DNN models in life-critical applications, e.g., vision-based self-driving or image-based medical diagnosis.

It was shown by Nguyen et al. (2015) (Nguyen et al., 2015) that DNN classifiers can be easily fooled by OOD data, and an evolutionarily algorithm was used to generate OOD samples such that DNN classifiers had high output confidence on these samples. Since then, many methods have been proposed for OOD detection using classifiers or encoders (Hendrycks & Gimpel, 2017; Hendrycks et al., 2019; Liang et al., 2018; Lee et al., 2018b;a; Alemi et al., 2018; Hendrycks & Gimpel, 2017). For instance, Hendrycks et al. (Hendrycks & Gimpel, 2017) show that a classifier's prediction probabilities of OOD examples tend to be more uniform, and therefore the maximum predicted class probability from the softmax layer was used for OOD detection. Regardless of the details of these methods, every method needs a classifier or an encoder, which takes an image x as input and compresses it into a vector $z$ in the laten space; after some further transform, $z$ is converted to an OOD detection score $\tau$. This computing process can be expressed as: $z = f(x)$ and $\tau = d(z)$. To perform OOD detection, a detection threshold needs to be specified, and then $x$ is OOD if $\tau$ is smaller/larger than the threshold. For the evaluation of OOD detection methods, (Hendrycks & Gimpel, 2017), an OOD detector is usually trained on a dataset (e.g. Fashion-MNIST as in-distribution) and then it is tested on another dataset (e.g. MNIST as OOD).

As will be shown in this study, the above mentioned classification-based OOD detection methods are practically breakable. As an example (more details in Section 3), we used the Resnet-18 model (He et al., 2016) pre-trained on the ImageNet dataset. Let $x_{in}$ denote a 224×224×3 image (in-distribution sample) in ImageNet and $x_{out}$ denote an OOD sample which could be any kinds of images (even random noises) not belonging to any category in ImageNet. Let $z$ denote the 512-dimensional feature vector in Resnet-18, which is the input to the last fully-connected linear layer before the softmax operation. Thus, we have $z_{in} = f(x_{in})$ and $z_{out} = f(x_{out})$. In Fig. 1, $x_{in}$

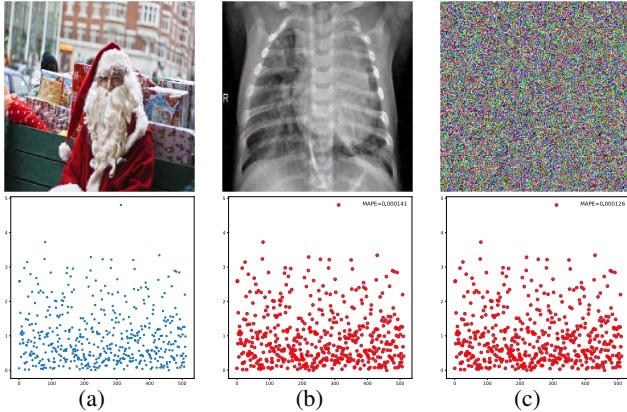

Figure 1: The 1st column shows the image of Santa Claus $x_{in}$ and the scatter plot of $z_{in}$ using blue dots. The 2nd column shows a chest x-ray image $x_{out}$ and the scatter plot of $z_{out}$ (red circles) and $z_{in}$ (blue). The 3rd column shows a random image $x_{out}$, and the scatter plot of $z_{out}$ (red) and $z_{in}$ (blue).

is the image of Santa Claus, and $x_{out}$ could be a chest x-ray image or a random-noise image, and "surprisingly", $z_{out} \cong z_{in}$ which renders OOD detection score to be useless: $d(z_{out}) \cong d(z_{in})$.

In Section 2, we will introduce an algorithm to generate OOD samples such that $z_{out} \cong z_{in}$. In Section 3, we will show the evaluation results on publicly available datasets, including ImageNet subset, GTSRB, OCT, and COVID-19 CT. Since some generative models (e.g. Glow (Kingma & Dhariwal, 2018)) can approximate the distribution of training samples (i.e. $p(x_{in})$), likelihood-based generative models were utilized for OOD detection (Nalisnick et al., 2019). It has been shown that likelihoods derived from generative models may not distinguish between OOD and training samples (Nalisnick et al., 2019; Ren et al., 2019; Choi et al., 2018), and a fix to the problem could be using likelihood ratio instead of raw likelihood score (Serrà et al., 2019). Although not the main focus of this study, we will show that the OOD sample's likelihood score from the Glow model (Kingma & Dhariwal, 2018; Serrà et al., 2019) can be arbitrarily manipulated by our algorithm (Section 2.1) such that the output probability $p(x_{in}) \cong p(x_{out})$, which further diminishes the effectiveness of any Glow likelihood-based detection methods.

## 2 METHODOLOGY

### 2.1 OOD ATTACK ON DNN ENCODER

We introduce an algorithm to perform OOD attack on a DNN encoder $z = f(x)$ which takes an image $x$ as input and transforms it into a feature vector $z$ in a latent space. Preprocessing on $x$ can be considered as the very first layer inside of the model $f(x)$. The algorithm needs a weak assumption that $f(x)$ is sub-differentiable. A CNN classifier can be considered a composition of a feature encoder $z = f(x)$ and a feature classifier $p = g(z)$ where $p$ is the softmax probability distribution over multiple classes.

Let's consider an in-distribution sample $x_{in}$ and an OOD sample $x'_{out}$, and apply the model: $z_{in} = f(x_{in})$ and $z'_{out} = f(x'_{out})$. Usually, $z'_{out} \neq z_{in}$. However, if we add a relatively small amount of noise $\delta$ to $x'_{out}$, then it could be possible that $f(x'_{out} + \delta) = z_{in}$ and $x'_{out} + \delta$ is still OOD. This idea is realized in Algorithm 1, OOD Attack on DNN Encoder.

The clip operation in Algorithm 1 is very important: it will limit the difference between $x_{out}$ and $x'_{out}$ so that $x_{out}$ may be OOD. The algorithm is inspired by the method called projected gradient descent (PGD) (Kurakin et al., 2016; Madry et al., 2018) which is used for adversarial attacks. We note that the term "adversarial attack" usually refers to adding a small perturbation to a clean sample $x$ in a dataset such that a classifier will incorrectly-classify the noisy sample while being

---

**Algorithm 1** OOD Attack on DNN Encoder

---

**Input:** An in-distribution sample $x_{in}$ in a dataset. An OOD sample $x'_{out}$ not similar to any sample in the dataset. $f$, the neural network feature encoder. $\epsilon$, the maximum perturbation measured by Lp norm. $N$, the total number of iterations. $\alpha$ the learning rate of the optimizer.

**Output:** an OOD sample $x_{out}$ s.t. $f(x_{out}) \cong f(x_{in})$

    **Process:**

1: Generate a random noise $\xi$ with $||\xi|| \leq \epsilon$

2: Initialize $x_{out} = x'_{out} + \xi$

3: Setup loss $J(x_{out}) = ||f(x_{out}) - f(x_{in})||^2$ (L2 norm)

4: **for** $n$ from 1 to $N$ **do**

5:      $x_{out} \leftarrow clip(x_{out} - \alpha \cdot h(J'(x_{out})))$, where $J'(x) = \partial J / \partial x$.

6: **end for**

    **Note:** The clip operation ensures that $||x_{out} - x'_{out}||_p \leq \epsilon$. The clip operation also ensures that pixel values stay within the feasible range (e.g. 0 to 1). If L-inf norm is used, $h(J')$ is the sign function; and if L2 norm is used, $h(J')$ is a function that normalizes $J'$ by its L2 norm. Adamax optimizer is used in the implementation

---

able to correctly-classify the original clean sample $x$. Thus, OOD attack and adversarial attack are completely different things.

In practice, the Algorithm 1 can be repeated many times to find the best solution. Random initialization is performed in line-1 and line-2 of the algorithm process. By adding initial random noise $\xi$ to $x'_{out}$, the algorithm will have a better chance to avoid local minima caused by a bad initialization.

## 2.2 DIMENSIONALITY REDUCTION AND OOD ATTACK

Recall that in a classification-based OOD Detection approach, a DNN encoder transforms the input to a feature vector, i.e., $z = f(x)$, and an OOD detection score is computed by another transform on $z$, i.e., and $\tau = d(z)$. If $z_{out} \cong z_{in}$, then $d(z_{out}) \cong d(z_{in})$ which breaks the OOD detector regardless of the transform $d$. Usually, a DNN encoder makes dimensionality reduction: the dimension of $z$ is significantly smaller than the dimension of $x$. In the example shown in Fig. 1, $z$ is a 512-dimensional feature vector ($dim(z) = 512$) in Resnet-18, and the dimension of $x$ is 150528 ($224 \times 224 \times 3$).

**Dimensionality reduction in an encoder provides the opportunity for the existence of the mapping of OOD and in-distribution samples to the same locations in the latent space.** This is simply because the vectors in a lower-dimensional space cannot represent all of the vectors/objects in a higher-dimensional space, which is the Pigeonhole Principle. Let's do an analysis on the Resnet-18 example in Fig. 1. A pixel of the color image $x$ has 8-bits. In the 150528-dimension discrete input space, there are $256^{224 \times 224 \times 3}$ different images/vectors, which defines the size of the input space. float32 data type is usually used in computation, a float32 variable can roughly represent $2^{32}$ unique real numbers. Thus, in the 512-dimensional latent space, there are $2^{32 \times 512}$ unique vectors/objects, which defines the size of the latent space. The ratio is $\left( \frac{2^{32 \times 512}}{256^{224 \times 224 \times 3}} \right) \ll 1$, and it shows that the latent space is significantly smaller than the input space. Thus, for some sample $x$ in the dataset, we can find another sample $x'$ such that $f(x') = f(x)$ as long as $dim(z) < dim(x)$. A question arises: will the $x'$ be in-distribution or OOD? To answer this question, let's partition the input discrete space $\Omega$ into two disjoint regions ($\Omega = \Omega_{in} \cup \Omega_{out}$), $\Omega_{in}$ of in-distribution samples and $\Omega_{out}$ of OOD samples. $|\Omega|$ denotes the size of $\Omega$. Usually, the training set is only a subset of $\Omega_{in}$, and the size of $\Omega_{out}$ is significantly larger than the size of $\Omega_{in}$. For example, if $\Omega_{in}$ is ImageNet, then $\Omega_{out}$ contains medical images, noise images, and other weird images. If $\Omega_{in}$ contains human face images, then $\Omega_{out}$ contains non-face images and then $|\Omega_{in}| \ll |\Omega_{out}|$. The latent space (z-space) is denoted by $\mathcal{F}$ and partitioned into two subspaces: $\mathcal{F} = \mathcal{F}_{in} \cup \mathcal{F}_{out}$. An encoder is applied such that $\Omega_{in} \rightarrow \mathcal{F}_{in}$ and $\Omega_{out} \rightarrow \mathcal{F}_{out}$. If there is overlap $\mathcal{F}_{in} \cap \mathcal{F}_{out} \neq \emptyset$, then the encoder is **vulnerable** to OOD attack. Usually, the encoder is a part of a classifier trained to classify in-distribution samples into different classes, and therefore the encoder **cannot guarantee that** there is no overlap between $\mathcal{F}_{in}$ and $\mathcal{F}_{out}$. What is the size of $\mathcal{F}_{in} \cap \mathcal{F}_{out}$ or what is the probability $P(|\mathcal{F}_{in} \cap \mathcal{F}_{out}| \geq a)$? While it is hard to calculate it for an arbitrary encoder and dataset, we can do a worst-case-scenario analy-

sis. Assuming that every OOD sample is i.i.d. mapped to the latent space with a uniform distribution over a number of $|\mathcal{F}|$ spots, then the probability of OOD samples covering the entire latent space is $P\left(\mathcal{F}_{out} = \mathcal{F}\right) = |\mathcal{F}|! \times Stirling(|\Omega_{out}|, |\mathcal{F}|)/|\mathcal{F}|^{|\Omega_{out}|} \to 1$ as $|\mathcal{F}|/|\Omega_{out}| \to 0$, where $Stirling$ is the Stirling number of the second kind. Noting that $|\mathcal{F}|/|\Omega_{out}| = \frac{2^{32 \times 512}}{256^{224 \times 224 \times 3} - 1.4 \times 10^7} \approx 0$ and $1.4 \times 10^7$ being the number of samples in ImageNet, then it could be true that almost (with probability close to 1) the entire latent space of Resnet-18 is covered by the $z$ vectors of OOD samples.

Next, we discuss how to construct OOD samples to fool neural networks. First, let's take a look at one-layer linear network: $z = Wx$, and make notations: an in-distribution input $x \in \mathcal{R}^M$, latent code $z \in \mathcal{R}^K$ and $K \ll M$. $W$ is a $K \times M$ matrix, and $rank\left(W\right) \leq K$. The null space of $W$ is $\Omega_{null} = \{\eta;\ W\eta = 0\}$. Now, let's take out the basis vectors of this space, $\eta_1, \eta_2, \ldots, \eta_{M-K}$, and compute $x' = \sum_i \lambda_i \eta_i + x$ where $\lambda_i$ is a non-zero scalar. Obviously, $z' = Wx' = z$. We can set the magnitude of the "noise" $\sum_i \lambda_i \eta_i$ to be arbitrarily large such that $x'$ will look like garbage and become OOD, which is another explanation of the existence of OOD samples. Then, we can try to apply this attack method to multi-layer neural network. If the neural network only uses ReLU activation, then the input-output relationship can be exactly expressed as a piecewise-linear mapping (Ding et al., 2020), a similar approach can be applied layer by layer. If ReLU is not used, a new method is needed. We note that the filter bank of a convolution layer can be converted to a weight matrix. We have examined the state-of-the-art CNN models that are pre-trained on ImageNet and available in Pytorch, and dimensionality reduction is performed in most of the layers (except 1 or 2 layers near the input), i.e. $|\mathcal{F}| \leq |\Omega_{in}| \ll |\Omega_{out}|$. Instead of constructing an OOD sample by adding perturbations to an in-distribution sample, in Algorithm-1, we construct an OOD sample paired with an in-distribution sample by starting from an initial sample that is OOD.

Could an encoder be made robust to the OOD attack by including OOD samples in training set for supervised binary classification: in vs out? Usually $|\Omega_{in}| \ll |\Omega_{out}|$ and we will have to collect and label "enough" samples in $\Omega_{out}$, which is infeasible considering the large size of $\Omega_{out} \approx \Omega$. As a comparison, to enhance DNN classifier robustness against adversarial noises, it is very effective to include noisy samples in the training set, i.e. $\Omega_{in} = \Omega_{in\_clean} \cup \Omega_{in\_noisy}$. It is known as adversarial training (Goodfellow et al., 2018) and computationally feasible as $|\Omega_{in\_noisy}| \ll |\Omega_{out}|$.

## 2.3 PROBLEM OF GLOW LIKELIHOOD-BASED OOD DETECTION

Generative models have been developed to approximate the training data distribution. Glow (Kingma & Dhariwal, 2018) is one of these models, and it has a very special property: it is bijective and the latent space dimension is the same as the input space dimension, i.e., no dimensionality reduction, which is the reason that we studied this model.

Several studies have found the problem of Glow-based OOD detection: likelihoods derived from Glow may not distinguish between OOD and training samples (Ren et al., 2019; Choi et al., 2018), and a possible fix to the issue could be using likelihood ratio (Serrà et al., 2019). In this study, we further show that negative log-likelihood (NLL) from the Glow model can be arbitrarily manipulated by our algorithm in which $f(x)$ denotes NLL. The results on CelebA face image dataset are in Section 3. We think the major reason causing Glow's vulnerability to OOD attack is that we do not have enough training data in high dimensional space. Glow is a mapping: $x_{in} \to z_{in} \to p(z_{in}) \to p(x_{in})$, the probability of $x_{in}$. For an OOD sample $x_{out}$, the mapping is $x_{out} \to z_{out} \to p(z_{out}) \to p(x_{out})$. Since the number of training samples is significantly smaller than the size of the space, there are a huge number of "holes" in the latent space (i.e., regions that no training samples are mapped to), and it is easy to put $z_{out}$ in one of these "holes" close to $z_{in}$ such that $p(z_{out}) \cong p(z_{in})$.

## 2.4 RECONSTRUCTION-BASED OOD DETECTION

Auto-encoder style OOD detection has been developed for anomaly detection (Chalapathy & Chawla, 2019; Cohen et al., 2019) based on reconstruction error. The data flow of an auto-encoder is $x \to z \to \hat{x}$ where $\hat{x}$ is the reconstruction of $x$. The OOD detection score can be the difference between $x$ and $\hat{x}$, e.g., the Lp distance $||x - \hat{x}||_p$ or Mahalanobis Distance. This type of method has two known issues. The first issue is that auto-encoder may well reconstruct OOD samples, i.e., $x_{out} \approx \hat{x}_{out}$. Thus, one needs to make sure it has large reconstruction errors on OOD samples, which can be done by limiting the capacity of auto-encoder or saturating it with in-distribution samples.

The second issue is that pixel-to-pixel distance is not a good measurement of image dissimilarity, especially for medical images. For example, $x$ could be a CT image of a heart and $\hat{x}$ could be the image of the same heart that deforms a little bit, but the pixel-to-pixel distance between $x$ and $\hat{x}$ can be very large. Thus, a robust image similarity measurement is needed.

Interestingly, the proposed OOD attack algorithm has no effect on this type of method. Let's consider the data flow: $x_{in} \rightarrow z_{in} \rightarrow \hat{x}_{in}$ and $x_{out} \rightarrow z_{out} \rightarrow \hat{x}_{out}$. If $z_{out} = z_{in}$, then $\hat{x}_{out} = \hat{x}_{in}$. Then, it is easy to find out that $x_{out}$ is OOD because $||x_{out} - \hat{x}_{out}||_p = ||x_{out} - \hat{x}_{in}||_p$ which is very large. Ironically, in this case, the attack algorithm helps to identify the OOD sample. In future work, we will evaluate the effectiveness of combining the proposed algorithm and auto-encoder for OOD detection.

## 3 Experiments

We applied the proposed algorithm to attack state-of-the-art DNN models on image datasets. For each in-distribution sample $x_{in}$, an OOD sample $x_{out}$ is generated by the algorithm. To measure attack strength, mean absolute percentage error is calculated by $MAPE(z_{out}) = mean(|z_{out} - z_{in}|)/max(|z_{in}|)$. Here, $z_{out} = f(x_{out})$ and $z_{in} = f(x_{in})$. $|z_{out} - z_{in}|$ is an error vector, and $mean(|z_{out} - z_{in}|)$ is the average error. $max(|z_{in}|)$ is the maximum absolute value in the vector $z_{in}$. We also applied the algorithm to attack the Glow model on CelebA dataset. In all of the evaluations, L2 norm was used in the proposed algorithm. Pytorch was used to implement the algorithm. Nvidia Titan V GPUs were used for model training and testing.

### 3.1 Evaluation on a Subset of ImageNet

ILSVRC2012 ImageNet has over 1 million images in 1000 classes. Given the limited computing power, it is impractical to test the algorithm on the whole dataset. Instead, we used a subset of 1000 images in 200 categories. The size of each image is 224×224×3. Two CNN models pretrained on the ImageNet were evaluated, which are Resnet-18 and Densenet-121 available in Pytorch.

Resnet-18 latent space has 512 dimensions. Since ImageNet covers a variety of natural and artificial objects, we choose medical images and random-noise images to make sure that $x'_{out}$ is indeed OOD. Using each of the three initial OOD samples (chest x-ray, lung-CT, and random noise to be $x'_{out}$), we generated 1000 OOD samples paired with the 1000 (in-distribution) samples in the dataset and calculated MAPE values. The three MAPE histograms are shown in Fig. 3. Most of the MAPE values are less than $0.1\%$.

We also evaluated another CNN, named Densenet-121, and obtained similar results. The latent space has 1024 dimensions. Again, using each of the three initial OOD samples, 1000 OOD samples are generated for the samples in the dataset, and then MAPE values are calculated. The three MAPE histograms are shown in Fig. 4. Most of the MAPE values are less than $0.1\%$, indicating strong OOD attack.

From the results in Fig.2 to Fig. 4, it can be seen that each of the two CNN models mapped significantly different OOD samples and in-distribution samples to almost the same locations in the latent space. Dimensionality reduction leads to the existence of such mapping, and our algorithm can find such OOD samples out. In other words, the mapping from input space to the latent space is many-to-one, not bijective. And therefore, it is almost guaranteed that such OOD samples exist and they can break any OOD detector $d$ that computes a detection score $d(z)$ only from the latent space (z-space). We tested a classical OOD detection method using the maximum of softmax output as detection score (Hendrycks & Gimpel, 2017). The results are shown in Table-1, and the AUROC scores are close to 0.5, showing that the method is unable to tell the difference between the 1000 OOD samples and 1000 in-distribution samples.

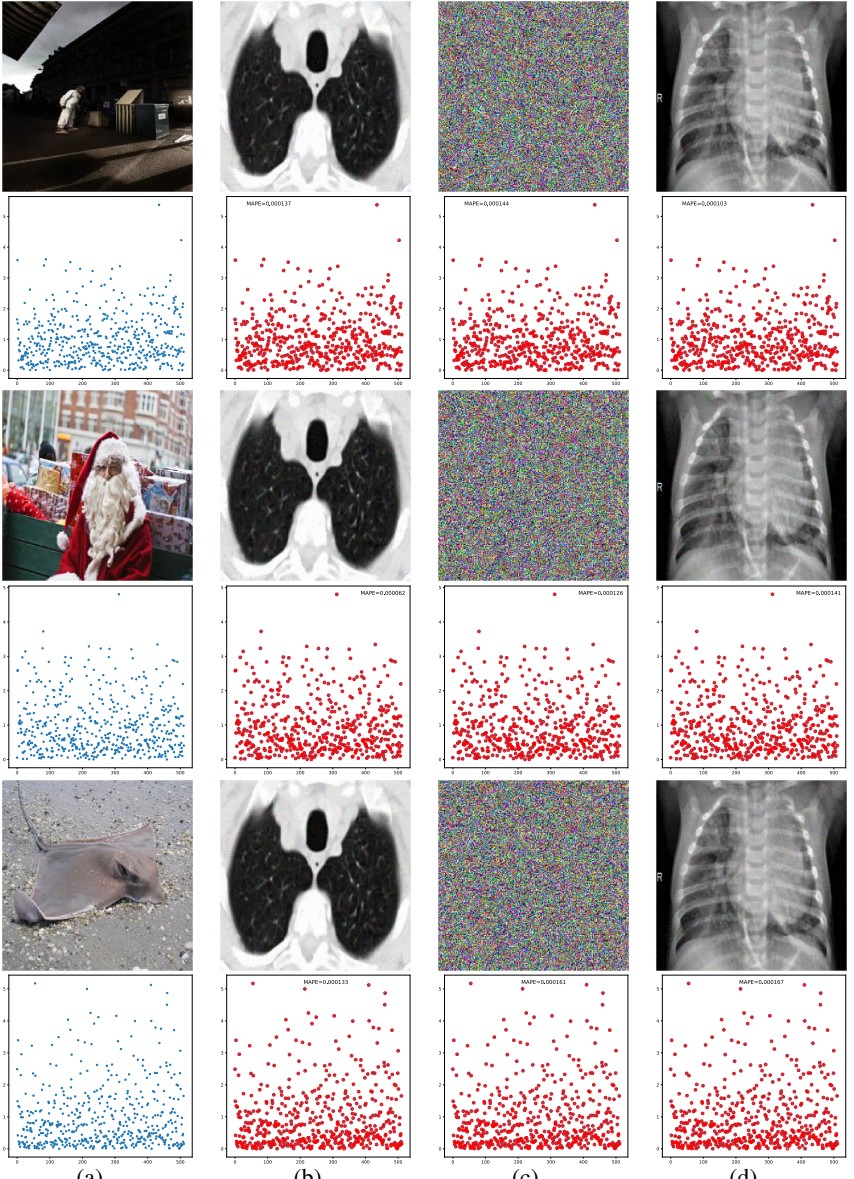

Figure 2: The 1st column shows three in-distribution samples (e.g. Santa Claus) $x_{in}$, and the corresponding scatter plots of $z_{in}$ (blue dots). The 2nd column shows OOD samples $x_{out}$ generated from a CT image $x'_{out}$, and the corresponding scatter plots of $z_{out}$ (red) and $z_{in}$ (blue). The 3rd column shows OOD samples $x_{out}$ generated from a random image $x'_{out}$, and the corresponding scatter plots of $z_{out}$ (red) and $z_{in}$ (blue). The 4th column shows OOD samples $x_{out}$ generated from a x-ray image $x'_{out}$, and the corresponding scatter plots of $z_{out}$ (red) and $z_{in}$ (blue). MAPE values are embedded in these scatter-plots. Please zoom-in for better visualization.

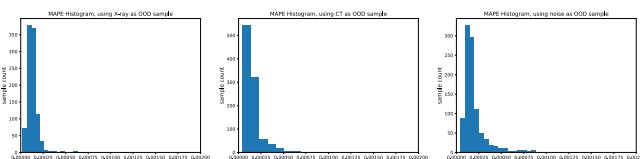

Figure 3: Left: the MAPE histogram using a chest x-ray as the initial OOD sample. Middle: the MAPE histogram using a lung CT image as the initial OOD sample. Right: the MAPE histogram using a random-noise image as the initial OOD sample. The results are from Resnet-18.

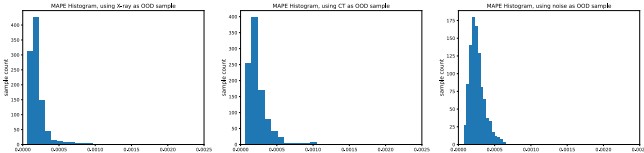

Figure 4: Left: MAPE histogram using a chest x-ray as the initial OOD sample. Middle: MAPE histogram using a lung CT image as the initial OOD sample. Right: MAPE histogram using a random-noise image as the initial OOD sample. The results are from Densenet-121.

Table 1: AUROC of two networks under OOD attack with each of x-ray, CT and noise as the initial OOD sample

|  | **x-ray** | **CT** | **random-noise** |
|---|---|---|---|
| Resnet-18 | 0.643 | 0.633 | 0.500 |
| Densenet-121 | 0.638 | 0.651 | 0.500 |

## 3.2 EVALUATION ON CELEBA DATASET

We tested the algorithm and the Glow model (Kingma & Dhariwal, 2018) on the CelebA dataset (human face images). The size of each image is 64×64×3. After training, the model was able to generate realistic face images. The model also outputs the negative log-likelihood (NLL) of the input sample, i.e., $NNL(x) = -log(p(x))$. By setting $f(x) = NNL(x)$, our algorithm can make $f(x_{out})$ to be close to 0 or very large to match any $f(x_{in})$, which renders NLL score useless for OOD detection. To demonstrate the effectiveness of our algorithm, we randomly selected 160 (in-distribution) samples in the dataset. We used a color spiral image as the initial OOD sample $x'_{out}$, and $NNL(x'_{out}) = 3.5268$. The distributions of $NLL(x_{in})$ from 160 in-distribution samples and $NLL(x_{out})$ from 160 corresponding OOD samples, as well as OOD sample images are shown in Fig. 5. The two distributions are almost identical. More examples of OOD samples are shown in Fig. 6. In each row of Fig. 6, although the images have different NLL scores, they look like each other.

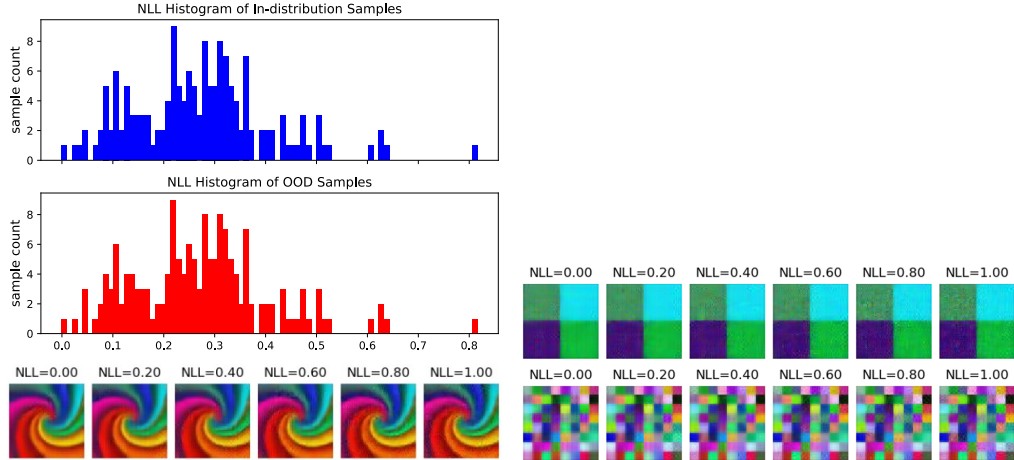

Figure 5: Top: NLL histogram (blue bars) of the in-distribution; samples Middle: NLL histogram (red bars) of OOD samples; Bottom: some OOD samples with NLL from 0 to 1. The initial OOD sample is a spiral image.

Figure 6: Top: OOD samples generated by using one 2×2 checkerbox image for initialization; Bottom: OOD samples generated by using one 8×8 checkerbox image for initialization.

**We have done more evaluations of our algorithm and OOD detection methods, please read the appendices.**

## 4    DISCUSSION

We hypothesized that dimensionality reduction in an encoder provides the opportunity for the existence of the mapping of OOD and in-distribution samples to the same locations in the latent space. We applied the OOD Attack algorithm to DNN classifiers on various datasets (see Appendices A and B), and the results (i.e. low MAPE values) confirmed our hypothesis. The results imply that classifier/encoder -based OOD detection methods may be vulnerable to the OOD attack.

By using our OOD Attack algorithm, we evaluated nine OOD detection methods (see Appendices C to J). The AUROC scores of these methods are close to 0.5 in our experiments, which means these methods could not distinguish between the in-distribution samples (e.g. CIFAR10) and the OOD samples generated by our algorithms. Our algorithm was unable to break a recent method named Certified Certain Uncertainty (Meinke & Hein, 2020), because this method utilizes Gaussian mixture models (GMMs) in the input space (note: no dimensionality reduction in GMMs). However, it is well known that GMMs have convergence issues for high dimensional data (e.g. medical images).

Compared to adversarial attacks and defenses, it is much more difficult to defend against OOD attacks. Adversarial attacks and OOD attacks are doing completely different things to neural networks, although the attack algorithms may use similar optimization techniques. For image classification applications, an adversarial attack will add a small amount of noise to the input (clean) image, and the resulting noisy image is still human-recognizable. Therefore, the magnitudes of adversarial noises are constrained. For example, a noisy image of a panda is still an image of the panda. By the judgment of humans, the noisy image and the clean image are the images of the same object, and the two images should be classified into the same class. Compared to adversarial samples, OOD samples, which can be generated by our OOD Attack algorithm, have much more freedom (e.g. they can be random noises), as long as they do not look like in-distribution samples. Thus, OOD detection is very challenging.

We would like to point out that it is difficult to evaluate an OOD detector to "prove" that it can detect, say 90% of the OOD samples by experimentally testing it on $\Omega_{out}$ because $\Omega_{out}$ is too large to be tested on: $|\Omega_{in}| \ll |\Omega_{out}| \approx |\Omega|$. For example, if Fashion-MNIST is used as in-distribution, then MINST and Omniglot are usually as OOD, which is the "standard" approach in the literature. Clearly, MINST and Omniglot cannot cover $\Omega_{out}$ the space of OOD samples. If the image size is larger, then $|\Omega_{out}|$ becomes much larger. Could we design an evaluation method (experimental or analytical) that does not rely on OOD samples?

Before the OOD detection issue is fully resolved, for life-critical applications, any machine learning system that uses DNN classifiers should not make decisions independently and can only serve as assistants to humans. The OOD Attack algorithm and the experimental results can serve as a reference for the evaluation of new OOD detection methods.

We will release the code on GitHub when the paper is accepted. All figures are in high-resolution, please zoom in.

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

## A  APPENDIX

The parameters in each evaluation are listed.

1. Parameters for evaluation on ImageNet subset Attacking resnet-18: $\varepsilon = 5$, $N = 1e4$, $\alpha = \varepsilon/100$ with x-ray and CT; $\varepsilon = 20$, $N = 1e4$, $\alpha = \varepsilon/100$ with random noise. Attacking densnet-121: $\varepsilon = 5$, $N = 1e4$, $\alpha = \varepsilon/100$ with x-ray and CT; $\varepsilon = 30$, $N = 1e4$, $\alpha = \varepsilon/100$ with random noise. We inspected the OOD images from random noise: they are not recognizable to human vision.

2. Parameters for evaluation on OCT dataset $\varepsilon = 10$, $N = 1e4$, $\alpha = \varepsilon/100$ with retinal fundus photography image $\varepsilon = 20$, $N = 1e4$, $\alpha = \varepsilon/100$ with random noise.

3. Parameters for evaluation on COVID-19 CT dataset $\varepsilon = 20$, $N = 1e4$, $\alpha = \varepsilon/100$

4. Parameters for evaluation on GTSRB dataset $\varepsilon = 10$, $N = 1e4$, $\alpha = \varepsilon/100$

5. Parameters for Evaluation on CelebA Dataset $\varepsilon = 10$, $N = 1e4$, $\alpha = \varepsilon/100$

## B  APPENDIX

### B.1  EVALUATION ON OCT DATASET

We tested our algorithm and Resnet-18 on a retinal optical coherence tomography (OCT) dataset (Kermany et al., 2018), which has four classes. Each image is resized to 224×224. 1000 samples per class were randomly selected to obtain a training set of 4000 samples. The test set has 968 images. We modified Resnet-18 for this four-class classification task. The latent space has 512 dimensions. After training, the Resnet-18 model achieved a classification accuracy $> 95\%$ on the test set.

We used two references images as the initial OOD sample $x'_{out}$. The first reference image is a grayscale retinal image converted from an RGB color retinal fundus photography image. Compared to this retinal fundus photography image, the OCT images have unique patterns of horizontal "white

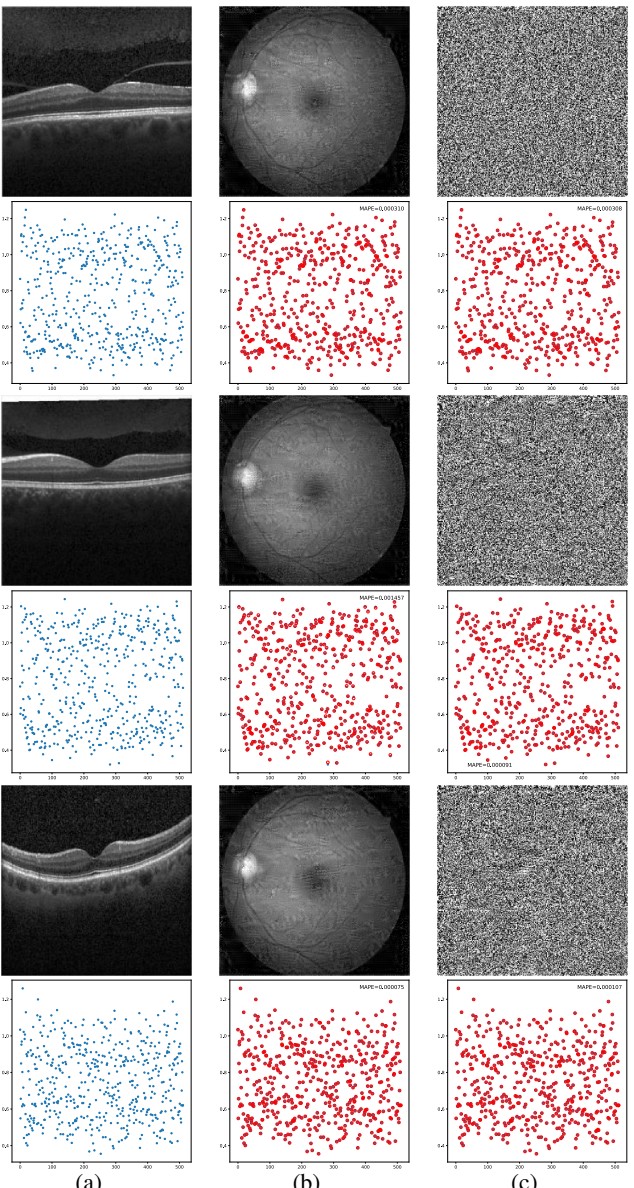

Figure 7: The 1st column shows in-distribution samples $x_{in}$ in the OCT dataset, and the scatter plots of $z_{in}$ (blue dots). The 2nd column shows OOD samples $x_{out}$ generated from a retinal fundus photography image $x'_{out}$, and the scatter plots of $z_{out}$ (red) and $z_{in}$ (blue). The 3rd column shows OOD samples $x_{out}$ generated from a random image $x'_{out}$, and the scatter plots of $z_{out}$ (red) and $z_{in}$ (blue). MAPE values are embedded in these scatter-plots. Please zoom-in for better visualization.

bands". We selected this OOD image by purpose: there may be a chance that both types of images are needed for retinal diagnosis. The second reference image is generated from random noises. Examples are shown in Fig. 7, and the two MAPE histograms are shown in Fig. 8. The results confirm that the algorithm can generate OOD samples (968) which are mapped by the DNN model to the locations of the in-distribution samples (968) in the latent space, i.e., $z_{out} \cong z_{in}$.

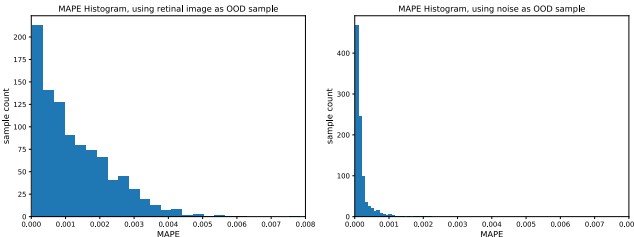

Figure 8: Left: MAPE histogram using a retinal fundus photography image as the initial OOD sample. Right: MAPE histogram using a random-noise image as the initial OOD sample. Please zoom-in for better visualization.

## B.2 EVALUATION ON COVID-19 CT DATASET

We also tested our algorithm and Resnet-18 on a public COVID-19 lung CT (2D) image dataset (Soares et al., 2020). It contains 1252 CT scans (2D images) that are positive for COVID-19 infection and 1230 CT scans (2D images) for patients non-infected by COVID-19, 2482 CT scans in total. From infected cases, we randomly selected 200 samples for testing, 30 for validation, and 1022 for training. From the uninfected cases, we randomly selected 200 for testing, 30 for validation and 1000 for training. Each image is resized to 224×224.

We modified the last layer of Resnet-18 for this binary classification task, infected vs uninfected. We also replaced batch normalization with instance normalization because it is known that batch normalization is not stable for small batch-size (Wu & He, 2018). The latent space still has 512 dimensions. We set batch-size to 32, the number of training epochs to 100, and used AdamW optimizer with the default parameters. After training, the model achieved a classification accuracy $> 95\%$ on test set.

We used two reference images as the initial OOD sample $x'_{out}$, a chest x-ray image, and a random-noise image. The two MAPE histograms are shown in Fig. 9 that most of the MAPE values are less than $0.1\%$. The results also confirm that the algorithm can generate OOD samples (400) which are mapped by the DNN model to the locations of the in-distribution samples (400) in the latent space, i.e., $z_{out} \cong z_{in}$.

Examples are shown in Fig. 10. As reported in the previous studies (Shi et al., 2020), infected regions in the images have a unique pattern called ground-glass opacity. The CT images in the 1st and 3rd rows show COVID-19 infections with ground-glass opacity on the upper-left area. The CT image in the 5th row does not show any signs of infection. It can be seen that the random-noise images and the COVID-19 CT images have the same feature vectors in the latent space, which is astonishing.

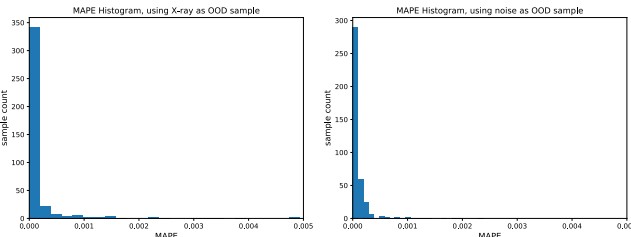

Figure 9: Left: MAPE histogram using chest x-ray image as the initial OOD sample. Right: MAPE histogram using a random-noise image as the initial OOD sample. Please zoom-in for better visualization.

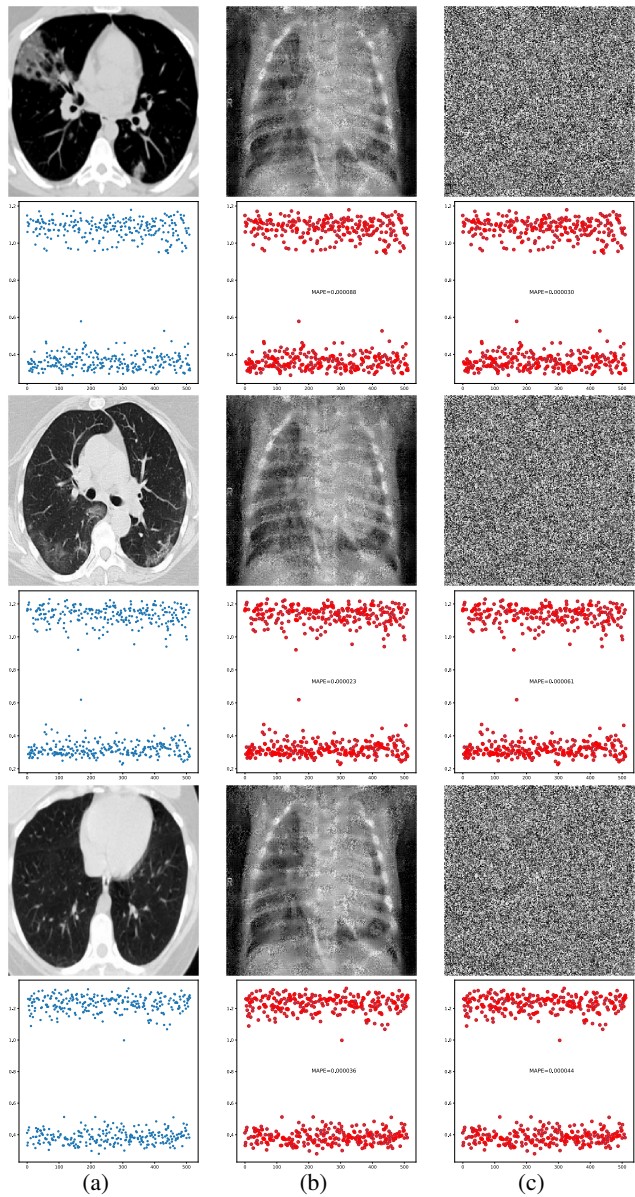

Figure 10: The 1st column shows in-distribution samples $x_{in}$ in the COVID-19 dataset, and the scatter plots of $z_{in}$ (blue dots). The 2nd column shows OOD samples $x_{out}$ generated from a chest x-ray image $x'_{out}$, and the scatter plots of $z_{out}$ (red) and $z_{in}$ (blue). The 3rd column shows OOD samples $x_{out}$ generated from a random image $x'_{out}$, and the scatter plots of $z_{out}$ (red) and $z_{in}$ (blue). MAPE values are embedded in these scatter-plots. Please zoom-in for better visualization.

## B.3 EVALUATION ON GTSRB TRAFFIC SIGN DATASET

We tested our algorithm and a state-of-the-art traffic sign classifier on the GTSRB dataset. The classifier is similar to the one in (Arcos-Garcia et al., 2018), which has a spatial-transformer network. The size of each image is 32×32×3. The latent space has 128 dimensions. After training, the classifier achieved over 99% accuracy on the test set. We used a random-noise image as the initial OOD sample $x'_{out}$ to generate 12630 OOD samples paired with the 12630 in-distribution samples in the test set. The MAPE histogram is shown in Fig. 11, in which most of the MAPE values are less than 0.1%. Examples are shown in Fig. 12.

It can be seen that $z_{out}$ of random-noise images are almost the same as $z_{in}$ of the stop sign, the speed limit sign, and the turning signs. Not only the classifier cannot tell the difference between a real traffic sign and a generated noise image, but also any detectors based on $z_{in}$ for OOD detection will fail. We note that adversarial robustness of traffic sign classifiers has been studied (Eykholt et al., 2018), and after adding adversarial noises to the traffic sign images, the noisy images are still recognizable. OOD noises and adversarial noises are very different (discussed in Sections 2.1 and 2.2). Thus, it would be wise to disable any vision-based auto-pilot in your self-driving cars today until this issue is resolved.

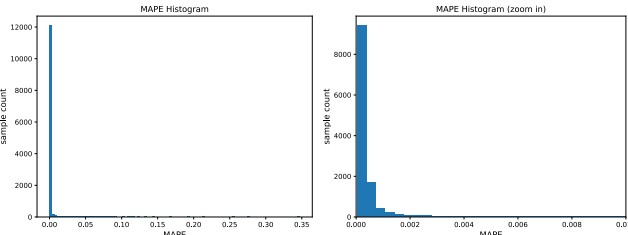

Figure 11: Left: MAPE histogram. Right: zoom-in view of the histogram

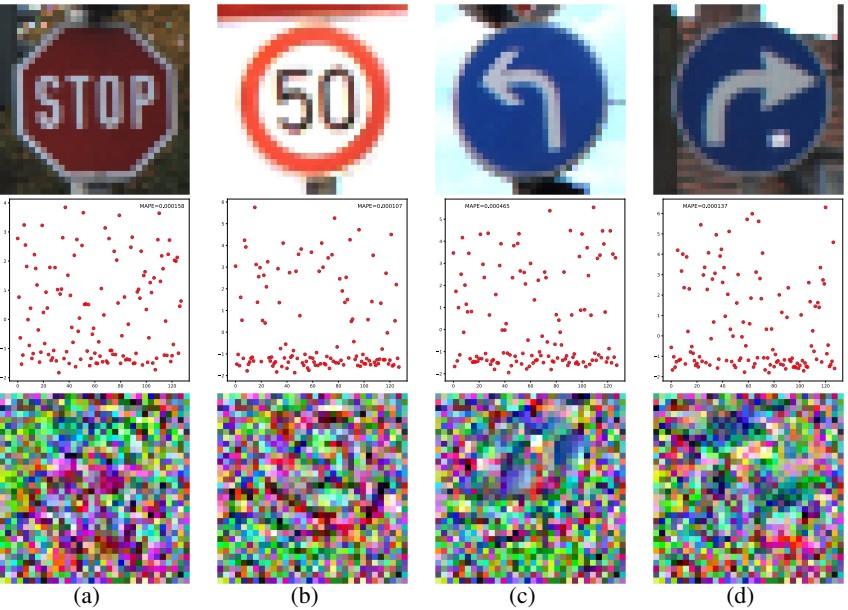

Figure 12: The 1st row shows four traffic sign images. The 3rd row shows the generated OOD images. The 2nd row shows the scatter-plots of $z_{out}$ (red) and $z_{in}$ (blue). MAPE values are embedded in these scatter-plots. Please zoom-in for better visualization.

## C APPENDIX

We applied our OOD Attack algorithm to test the OOD detection method named ODIN (Liang et al., 2018).

### C.1 SUMMARY OF THE ODIN METHOD

The method does temperature scaling on the logits (input to softmax) by logits/$T$, which is the Eq.(1) in the ODIN paper. The temperature $T$ could be in the range of 1 to 1000. The ODIN method also does input preprocessing, which is the Eq.(2) in the ODIN paper. For preprocessing, the perturbation

magnitude ($PM$) could be in the range of 0 to 0.004. The OOD score is defined to be the maximum of the softmax outputs from a neural network, given the preprocessed input. An OOD sample is expected to have a low OOD score.

### C.2 EVALUATION ON CIFAR10

Wide residual network with depth 28 and widen factor 10 is used in the ODIN paper. After training for 200 epochs, the model achieved the classification accuracy of 94.71 on CIFAR10 test set.

In our algorithm, we set $f(x)$ to be the logits output from the model, given the preprocessed input. For the CIFAR10 dataset, the logits output contains 10 elements, which is significant dimensionality reduction compared to the size of an input color image: $32 \times 32 \times 3$. In our algorithm, the parameters are $\varepsilon = 10, \alpha = \varepsilon/100, N = 100$. The initial OOD sample is a random noise image. For every sample in the CIFAR10 test set, the algorithm generated an OOD sample to match the logits output. The generated OOD samples look like random noises. The OOD scores of these samples were calculated by the ODIN method.

The results are reported in Table 2. Fig. 13 shows the OOD score histograms of the in-distribution and OOD samples when $T$=1000 and $PM$=0.001. When $T = 1$ and $PM = 0$, ODIN becomes the Baseline method (Hendrycks & Gimpel, 2017).

Table 2: AUROC scores of ODIN on CIFAR10 vs OOD

|  | T=1 | T=10 | T=100 | T=1000 |
|---|---|---|---|---|
| PM=0 | 0.500 | 0.500 | 0.500 | 0.500 |
| PM=0.001 | 0.500 | 0.500 | 0.500 | 0.500 |
| PM=0.002 | 0.500 | 0.500 | 0.500 | 0.500 |
| PM=0.004 | 0.500 | 0.500 | 0.500 | 0.500 |

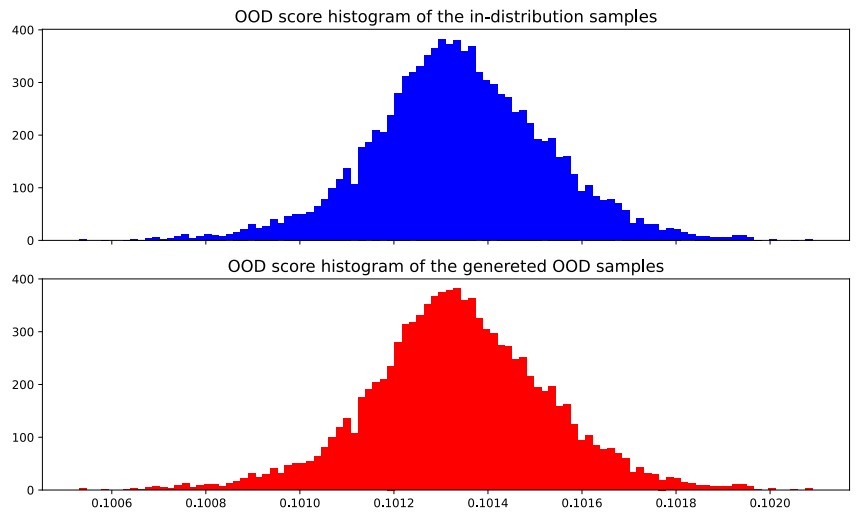

Figure 13: the OOD score histograms of the in-distribution (blue) and OOD (red) samples when T=1000 and PM=0.001

## D APPENDIX

We applied our OOD Attack algorithm to test the OOD detection method named Mahalanobis (Lee et al., 2018b).

### D.1 Summary of the Mahalanobis method

The method extracts the feature maps from multiple layers of a neural network and applies average pooling per channel to reduce each feature map into a 1D feature vector. Then, Mahalanobis distance is calculated between a feature vector and the corresponding mean vector. The distance values from all of the feature vectors are linearly combined together to produce a single distance, i.e., the OOD score. The OOD score of an OOD sample is expected to be large. To further improve the performance, the method does input preprocessing with a given perturbation magnitude (PM), and then OOD score of the preprocessed input is obtained. The weights to combine the Mahalanobis distances from multiple layers could be determined on a validation set of OOD samples. In practice, it is impossible to obtain such a validation set. In our evaluation, we simply take the average of the distance values, which gives the OOD score.

Although the feature maps from multiple layers are utilized, the method still does dimensionality reduction to those feature maps (e.g. averaging). Therefore, the method is breakable by our OOD Attack algorithm.

### D.2 Evaluation on CIFAR10 and CIFAR100

The neural network model is a residual network named Resnet34 in the Mahalanobis paper, and by changing the number of outputs, it can be used for CIFAR10 and CIFAR100. We used the pre-trained models that are available online at https://github.com/pokaxpoka/deep_Mahalanobis_detector/. The layers used for feature extraction, are exactly the same as those in the source code of the method.

In our algorithm, we have two different settings for $f(x)$: (1) it can be the OOD score, and (2) it can be the concatenation of the feature vectors, given the original (not preprocessed) input. We did experiments with the two settings. In our algorithm, the parameters are $\varepsilon = 10, \alpha = \varepsilon/100, N = 1000$ for all experiments. The initial OOD sample is a random noise image. For every sample in the test set, the algorithm generated an OOD sample to match the corresponding output. The generated OOD samples look like random noises. The OOD scores of these samples were calculated by the Mahalanobis method.

The results on the two datasets are reported in Table 3 and Table 4. Fig. 14 shows the OOD score histograms of the in-distribution and OOD samples, when the in-distribution dataset is CIFAR10, PM=0.01, and $f(x)$ = OOD score.

Table 3: AUROC scores of Mahalanobis on CIFAR10 vs OOD

|  | $f(x)$ = OOD score of $x$ | $f(x)$ = feature concatenation |
|---|---|---|
| PM=0 | 0.500 | 0.467 |
| PM=0.01 | 0.500 | 0.179 |

Table 4: AUROC scores of Mahalanobis on CIFAR100 vs OOD

|  | $f(x)$ = OOD score of $x$ | $f(x)$ = feature concatenation |
|---|---|---|
| PM=0 | 0.500 | 0.604 |
| PM=0.01 | 0.500 | 0.377 |

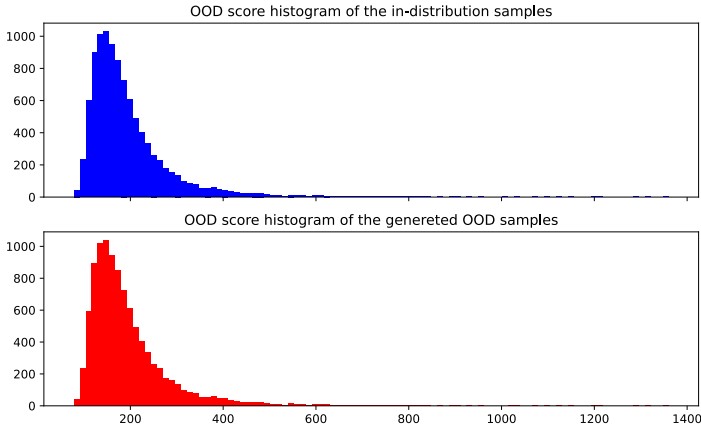

Figure 14: The OOD score histograms of the in-distribution (blue) and OOD (red) samples. The in-distribution dataset is CIFAR10, PM=0.01, and $f(x)$ = OOD score of $x$.

Fig. 15 shows the OOD score histograms of the in-distribution and OOD samples, when the in-distribution dataset is CIFAR10, PM=0.01, and $f(x)$ = feature concatenation. It can be seen that the OOD samples have smaller distances, which is caused by input preprocessing.

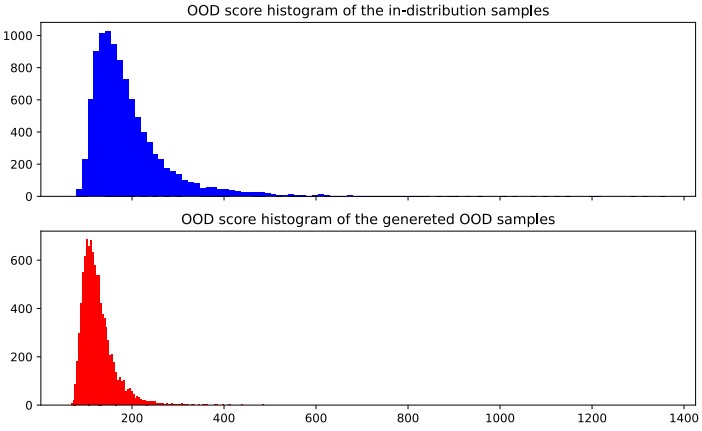

Figure 15: The OOD score histograms of the in-distribution (blue) and OOD (red) samples. The in-distribution dataset is CIFAR10, PM=0.01, and $f(x)$ = feature concatenation.

# E APPENDIX

We applied our OOD Attack algorithm to test the OOD detection method named Outlier Exposure (Hendrycks et al., 2019).

## E.1 SUMMARY OF THE OUTLIER EXPOSURE METHOD

The method trains a neural network on not only the standard training set (i.e. in distribution) but also an auxiliary dataset of outliers (i.e. OOD samples). In the paper, it states that the OOD score is defined to be the cross-entropy between a uniform distribution and the softmax-output distribution.

In the actual implementation (i.e. source code of the method), the OOD score is defined to be the average of the logits minus the logsumexp of the logits. In the evaluation, we used the actual implementation in the source code.

### E.2 EVALUATION ON SVHN, CIFAR10 AND CIFAR100

Wide residual networks are used in the Outlier Exposure paper. We downloaded the source code and pre-trained weights from https://github.com/hendrycks/outlier-exposure. The models were trained from scratch using Outlier Exposure, and they were named "oe_scratch" by the authors.

In our algorithm, we set $f(x)$ to be the logits (input to softmax) from each model. The parameters are $\varepsilon$=10, $\alpha = \varepsilon/100$, $N$=1e4 for all experiments. The initial OOD sample is a random noise image. For every sample in the test set, the algorithm generated an OOD sample to match the logits output. The generated OOD samples look like random noises. The OOD scores of these samples were calculated by the Outlier Exposure method.

The results are reported in Table 5. The OOD score histograms of the in-distribution and OOD samples are shown in Fig. 16, where the in-distribution dataset is CIFAR10.

Table 5: AUROC of Outlier Exposure on three datasets

| SVHN vs OOD | CIFAR10 vs OOD | CIFAR100 vs OOD |
|---|---|---|
| 0.500 | 0.500 | 0.500 |

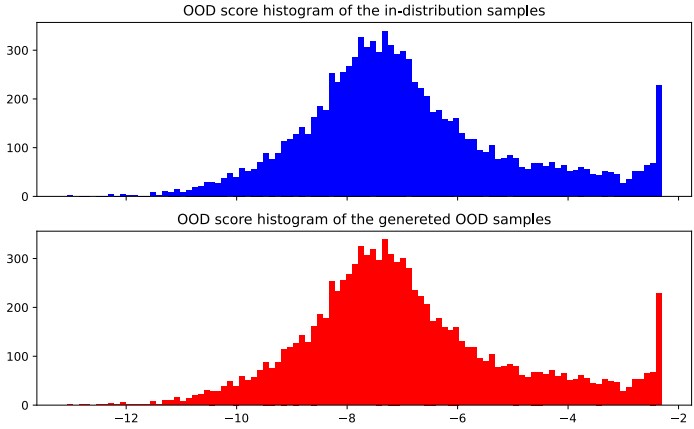

Figure 16: The OOD score histograms of the in-distribution (blue) and OOD (red) samples. The in-distribution dataset is CIFAR10.

## F APPENDIX

We applied our OOD Attack algorithm to test the OOD detection method named Deep Ensemble (Lakshminarayanan et al., 2017).

### F.1 SUMMARY OF THE DEEP ENSEMBLE METHOD

Deep Ensemble is a collection of neural network models working together for a classification task. The output of a Deep Ensemble is a probability distribution across the classes, which is the average of the probability/softmax outputs of individual models. In the experiments, the number of models is 5 in a Deep Ensemble. To further improve performance, adversarial training is applied to the

models. The OOD score is defined to be the entropy of the probability distribution from the Deep Ensemble. The entropy is expected to be large for an OOD sample.

### F.2    EVALUATE ON CIFAR10

The authors of the Deep Ensemble method did not provide source code and trained models. Therefore, we used pre-trained models from a recent work on adversarial robustness (Ding et al., 2020), which presented a state-of-the-art adversarial training method. Six pre-trained models were downloaded from https://github.com/BorealisAI/mma_training/tree/master/trained_models. The names of the models are cifar10-L2-MMA-1.0-sd0, cifar10-L2-MMA-2.0-sd0, cifar10-L2-OMMA-1.0-sd0, cifar10-L2-OMMA-2.0-sd0, cifar10-Linf-MMA-12-sd0, cifar10-Linf-OMMA-12-sd0. The models were trained on CIFAR10 to be robust against adversarial noises in a large range. Classification accuracy of the ensemble on test set is 89.85%.

In our algorithm, we set $f(x)$ to be the concatenation of the logits from each of the six models. The parameters are $\varepsilon=10$, $\alpha=\varepsilon/100$, $N=1e4$ for all experiments. The initial OOD sample is a random noise image. For every sample in the test set, the algorithm generated an OOD sample to match the logits output. The generated OOD samples look like random noises. The OOD scores of these samples were calculated by the Deep Ensemble method.

The AUROC of the Deep Ensemble method is 0.500 on CIFAR10 vs OOD. The OOD score histograms of the in-distribution and OOD samples are shown in Fig. 17.

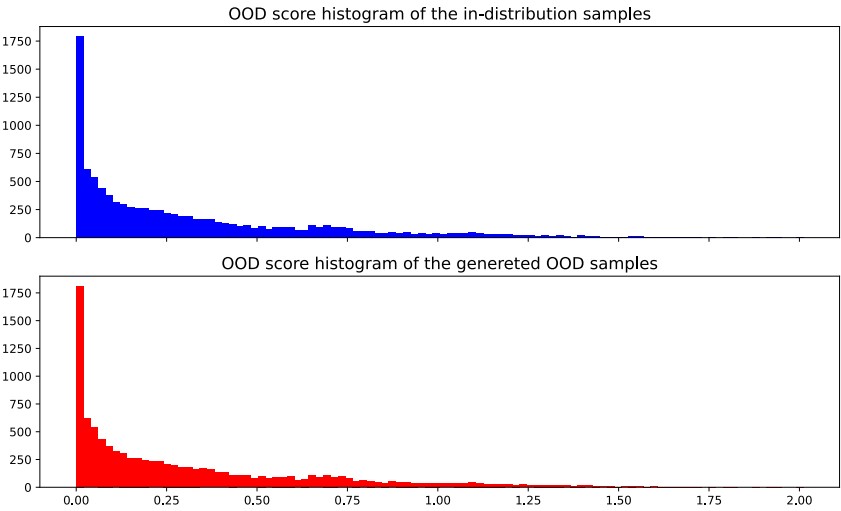

Figure 17: The OOD score histograms of the in-distribution (blue) and OOD (red) samples.

## G    APPENDIX

We applied our OOD Attack algorithm to test the OOD detection method that builds Confidence-Calibrated Classifiers (Lee et al., 2018a).

### G.1    SUMMARY OF THE OOD DETECTION METHOD

The method jointly trains a classification network and a generative neural network (i.e. GAN) that generates OOD samples for training the classification network. Given an input, the OOD score is defined to be the maximum of the softmax outputs from the classification network. The OOD score is expected to be low for an OOD sample.

### G.2 EVALUATION ON SVHN AND CIFAR10

The neural network model is VGG13, and the source code of the method is provided by the authors at https://github.com/alinlab/Confident_classifier. We downloaded the code and trained a VGG13 model with a GAN on SVHN and another VGG13 model with a GAN on CIFAR10 by using the parameters in the source code. VGG13 has a feature module and a classifier module.

In our algorithm, we set $f(x)$ to be the vector input to the classifier module of VGG13. The parameters are $\varepsilon$=10, $\alpha$= $\varepsilon$/100, $N$=1e4 for all experiments. The initial OOD sample is a random noise image. For every sample in the test set, the algorithm generated an OOD sample to match the vector input to the classifier module. The generated OOD samples look like random noises. The OOD scores of these samples were calculated by the OOD detection method.

The results are reported in Table 6 The OOD score histograms of the in-distribution and OOD samples are shown in Fig. 18, where the in-distribution dataset is CIFAR10.

Table 6: AUROC of the method on two datasets

| SVHN vs OOD | CIFAR10 vs OOD |
| --- | --- |
| 0.501 | 0.576 |

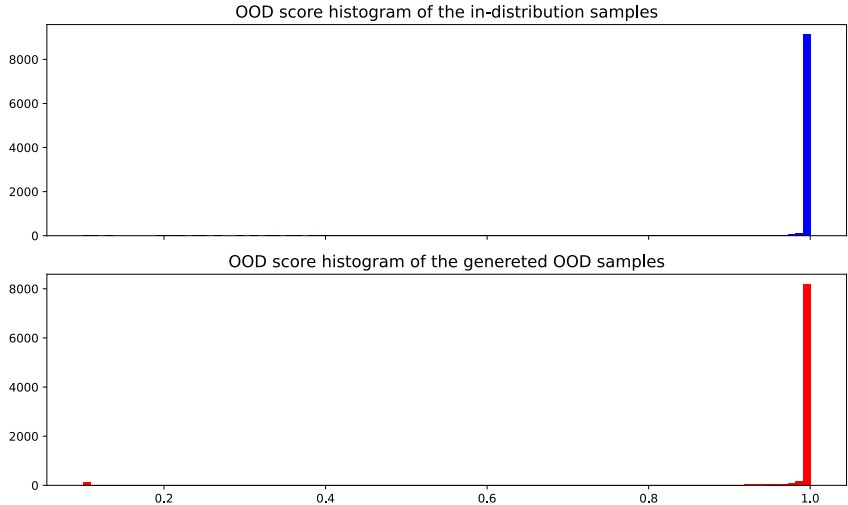

Figure 18: The OOD score histograms of the in-distribution (blue) and OOD (red) samples. The in-distribution dataset is CIFAR10.

## H APPENDIX

We applied our OOD Attack algorithm to test the OOD detection method named Gram (Sastry & Oore, 2019).

### H.1 SUMMARY OF THE GRAM METHOD

The method extracts the feature map $F_l$ from every layer of a network and then computes the p-th order Gram matrix $G_l^p = \left(F_l^p F_l^{pT}\right)^{1/p}$. Gram matrices with different p values from different layers are then used to compute the OOD score, which is named Total Deviation of the input sample. An OOD sample is expected to have a high OOD score.

## H.2  RESOLVING A NUMERICAL PROBLEM

The formula of the p-th order Gram matrix can be written as $A = (B)^{1/p}$. The Gram matrices caused gradients to be inf or nan during back-propagation in the OOD attack algorithm. To resolve this problem, we tried three tricks:

(a) use double precision (float64)

(b) rewrite $A = exp(\frac{1}{p}log(B + eps))$ where $eps$=1e-40

(c) use the equation in (b) to generate images during OOD attack and use the original equation $A = (B)^{1/p}$ to compute OOD scores.

The above tricks work for $p$ in the range of 1 to 5. For larger $p$, we still get numerical problems (inf or nan). As shown in Fig. 2 of the Gram paper, the method has already achieved better performance compared to the Mahalanobis method when the max value of $p$ is 5. Thus, we set the max value of $p$ to 5 in our experiments.

## H.3  EVALUATION ON CIFAR10 AND CIFAR100

The source code and pre-trained Resnet models are provided by the authors at https://github.com/VectorInstitute/gram-ood-detection

Due to the unique process of the method, it is very difficult to do parallel computing with mini-batches, and we have to set batch_size =1. The computing process is very time-consuming, and therefore we selected the first 500 samples in CIFAR10 test set and the first 500 samples in CI-FAR100 test set in our experiments.

In our algorithm, we set $f(x)$ to be the OOD score of $x$, and the parameters are $\varepsilon$=10, $\alpha$= $\varepsilon$/100, $N$=100. The initial OOD sample is a random noise image. For every in-distribution sample, the algorithm generated an OOD sample to match the OOD score. The generated OOD samples look like random noises. The OOD scores of these samples were calculated by the Gram method.

The results are reported in Table 7. The OOD score histograms of the in-distribution and OOD samples are shown in Fig. 19, where the in-distribution dataset is CIFAR10. The results show that the OOD score from the Gram method can be arbitrarily manipulated by our algorithm.

Table 7: AUROC of Gram on two datasets.

| CIFAR10 vs OOD | CIFAR100 vs OOD |
|---|---|
| 0.500 | 0.500 |

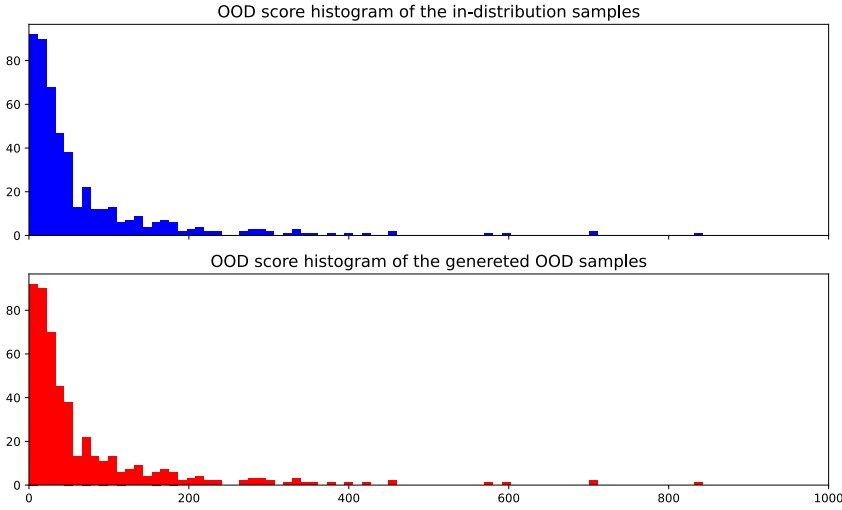

Figure 19: The OOD score histograms of the in-distribution (blue) and OOD (red) samples. The in-distribution dataset is CIFAR10.

# I  APPENDIX

We applied our OOD Attack algorithm to test the OOD detection method based on Glow (Serrà et al., 2019).

## I.1  SUMMARY OF THE OOD DETECTION METHOD

The method combines Glow negative log-likelihood (NLL) and input-complexity. PNG compression is used to compress the input image. The input-complexity, L, is measured by bits per dimension, where the "bits" refers to the number of bits of the compressed image, and the dimension is the total number of pixels per image. The OOD score is NLL – L.

## I.2  EVALUATE ON CELEBA

The source code of the Glow model is downloaded from https://github.com/rosinality/glow-pytorch, and we trained it from scratch on CelbaA dataset, in which the size of each face image is $64643$. After training, the model was able to generate realistic face images. For method evaluation, we randomly selected 160 samples in the dataset because the computation cost is very high.

In our algorithm, we set $f(x)$ to be the OOD score of $x$, and the parameters are $\varepsilon$=10, $\alpha$= $\varepsilon$/100, $N$=1e4. The initial OOD sample is a color spiral image. For every in-distribution sample, the algorithm generated an OOD sample to match the OOD score. The generated OOD samples look like color spirals, i.e., not face images. The OOD scores of these samples were calculated by the OOD detection method.

AUROC of the method is 0.500 on CelbaA vs OOD. The OOD score histograms of the in-distribution and OOD samples are shown in Fig. 20, The result indicates that NLL combined with input complexity can still be arbitrarily manipulated.

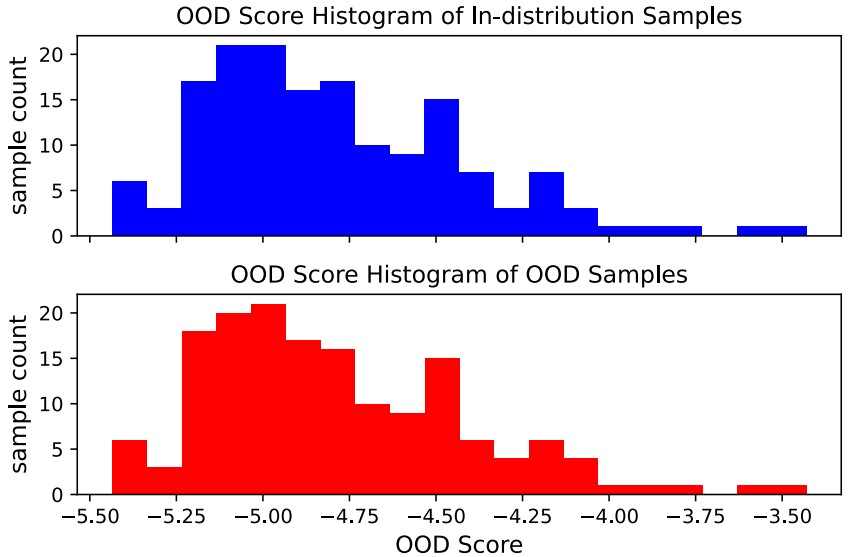

Figure 20: The OOD score histograms of the in-distribution (blue) and OOD (red) samples.

## J    APPENDIX

We applied our OOD Attack algorithm to test the OOD detection method using an energy-based model named JEM (Grathwohl et al., 2020).

### J.1    SUMMARY OF THE OOD DETECTION METHOD USING JEM

In the JEM paper, it was shown that a standard classifier can be trained to be an energy-based model (EBM). Using the EBM, three types of OOD scores can be obtained for an input sample, which are: (1) Log Likelihood $logp(x)$, (2) the maximum of softmax classification output, i.e. $max_y p(y|x)$, and (3) $-||\frac{\partial logp(x)}{\partial x}||_2$. An OOD sample is expected to have a low OOD score.

### J.2    EVALUATION ON CIFAR10

A wide residual network pretrained on CIFAR10 is available at https://github.com/wgrathwohl/JEM.

In our algorithm, we set $f(x)$ to be the logits (i.e. input to softmax for classification), and the parameters are $\varepsilon$=10, $\alpha$= $\varepsilon$/100, $N$=1e3. The initial OOD sample is a color spiral image. For every in-distribution sample, the algorithm generated an OOD sample to match the logits output. The generated OOD samples look very weird, not in any of the 10 classes of CIFAR10. The OOD scores of these samples were calculated by the OOD detection method.

We note that we tried to use random noises as initial OOD samples, but many generated images look like images in CIFAR10 dataset, and therefore, we used a color spiral image as the initial OOD sample.

The results are reported in Table 8. The OOD score histograms of the in-distribution and OOD samples are shown in Fig. 21, Fig. 22, and Fig. 23, We note that when using $-||\frac{\partial logp(x)}{\partial x}||_2$ as the OOD score, the AUROC is 0.203. One may think if we flip the sign of the OOD score, then AUROC will increase to 0.797. If we do so, then AUROC scores in the last row of Table 3 in the JEM paper will be close to 0 for the OOD detection experiments done by the authors.

Table 8: AUROC of the OOD detection method on CIFAR10 vs OOD

| OOD score | $logp(x)$ | $max_y p\left(y|x\right)$ | $-||\frac{\partial logp(x)}{\partial x}||_2$ |
|---|---|---|---|
| AUROC | 0.559 | 0.513 | 0.203 |

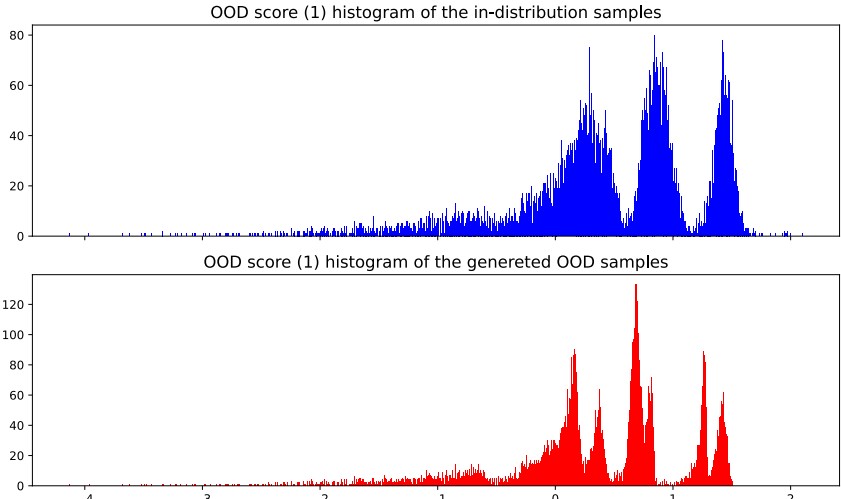

Figure 21: The OOD score ($logp(x)$) histograms of the in-distribution (blue) and OOD (red) samples.

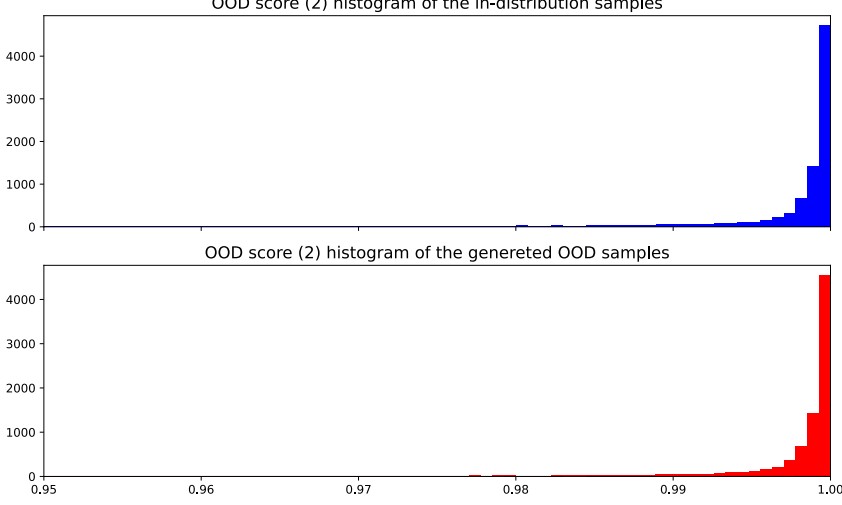

Figure 22: The OOD score ($max_y p\left(y|x\right)$) histograms of the in-distribution (blue) and OOD (red) samples.

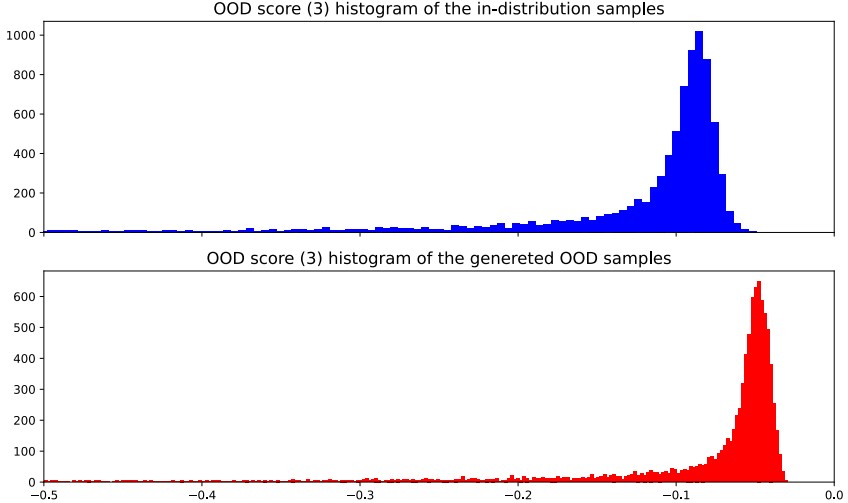

Figure 23: The OOD score $(-||\frac{\partial log p(x)}{\partial x}||_2)$ histograms of the in-distribution (blue) and OOD (red) samples.

Fig. 24 shows an example of the loss curve over 1000 iteration in the OOD attack algorithm.

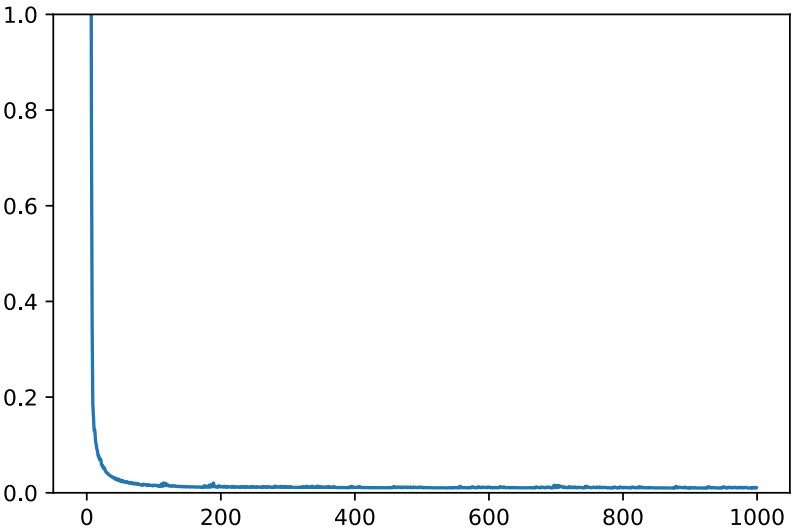

Figure 24: An example of the loss curve from the OOD Attack algorithm.

Fig. 25 shows some of the generated images, which look like the images of Frankenstein's monsters: randomly put some parts of objects together, twist/deform them, and then pour some paint on them. It may be difficult for neural networks to learn what is an object (e.g. airplane) just from images and class labels.

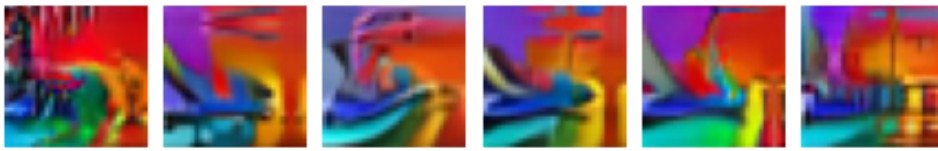

Figure 25: Examples of the generated images.

Energy-based models (EBMs), such as JEM, can generate OOD samples during training, which may explain why the OOD attack failed when the initial OOD sample was random noise. If we take a closer look at the sampling procedure (e.g. Langevin dynamics) and the objective function, it is easy to find out EBM training algorithm is trying to pull down the energy scores of positive (in-distribution) samples and pull up the energy scores of negative (OOD) samples (Du & Mordatch, 2019), which is similar to the basic idea of adversarial training. From this perspective, OOD detection using EBMs could be a promising direction if the computation cost is acceptable, and the challenge is how to train a neural network to learn what is an object?

