# OpenReview forum: "An Algorithm for Out-Of-Distribution Attack to Neural Network Encoder "
_ICLR.cc/2021/Conference — Reject_

### Official Review · AnonReviewer3 · 2020-10-19
**Experiments are not comprehensive - Claims are not validated - Organization needs improvement**

**Rating:** 3
**Confidence:** 4

**Review:**

Summary:
--------------
The paper presents a method that attacks existing out-of-distribution (OOD) detection methods. Most of the existing OOD detection methods perform detection using a latent representation. Main motivation of the paper is that the size of the latent representation is much smaller than the input images which results mapping both OOD and in-distribution images to the same place in the latent space and diminishing OOD detection performance. With this motivation, the proposed method perturbs input images to obtain an image whose latent representation is similar to the latent representation of an in-distribution image. Since such perturbations can be obtained for any OOD image, existing OOD detection algorithms fails distinguishing such OOD samples. The paper contains experiments on multiple dataset to demonstrate that the proposed method obtains a latent representation similar to the representation of an in-distribution image.

Comments:
----------------
1 - From the abstract, I infer that the paper has roughly 3 contributions: 1) the paper shows existing OOD detection methods are practically breakable, 2) Glow likelihood-based OOD detection is ineffective, and 3) present a simple theoretical solution with guaranteed performance for OOD detection. However, the 3rd contribution is never mentioned/introduced in the paper until Appendix A where they briefly presents 2 different implementations of an OOD detection method idea. I would not consider this as a contribution since there is no experimental evaluation showing the OOD detection performance of the idea. Also, since this contribution is mentioned in the abstract, I would expect seeing its description and the results in the main paper rather than the Appendix.

2 - Until the end of the Introduction, it is not very clear that the main contribution of the paper is a method that attacks OOD detection methods. I liked the motivating example in Figure 1, but the contribution can be given in a more clear way.

3 - In the last paragraph of page 2, it is mentioned that the clip operator can ensure x_out to be OOD after a small modification to x'_out. I found this statement quite vague. How this operation "ensures" that x_out to be OOD after the modification?

4 - In the last paragraph of Section 2.1, it is mentioned that adding initial random noise helps to avoid local minimum caused by a bad initialization. Is this something that you observed empirically? The contribution of the noise is not very clear to me and I think showing results with and without the noise would be very useful to demonstrate how the noise helps to avoid local minima. Also, it would be interesting to show how this loss evolves during the optimization.

5 - In Sec 2.2 - Each pixel in an 8-bit image can take 256 different values. So, for an image with size 224x224x3, there are 256^(224x224x3) possible images that can be generated; not 8^(224x224x3).

6 - I didn't quite understand the message in the last paragraph of Sec. 2.2. Why \Omega_in is split into \Omega_{in_clean} and \Omega_{in_noisy}? How this is used in the proposed method?

7 - I think that experimental evaluations are not comprehensive enough,
7.1. It is mentioned that implementing OOD detection methods are not necessary since Alg.1 already produces z_out ~= z_in. I don't agree with this since most of the SoTA methods do not rely on only a single representation but multiple representations at different layers [ref1, ref2, ref3]. Therefore, it is crucial to show the performance loss of these methods due to the proposed method.

[ref1] Erdil et al., Unsupervised out-of-distribution detection using kernel density estimation
[ref2] Lee et al., A Simple Unified Framework for Detecting Out-of-Distribution Samples and Adversarial Attacks
[ref3] Sastry et al., Detecting Out-of-Distribution Examples with Gram Matrices

7.2. Benchmark datasets that have been used in OOD detection paper are usually different than the ones used in the paper. Please see the references above. I wonder why the datasets choice are different than the standard benchmarks?

7.3. As mention in paragraph 3 of Sec. 4., there are other generative models that show promising results. However, I didn't quite understand comparison with these methods are "out of the scope" while Glow is relevant.

8. The parameters used in different experiments are different than each other. How are these parameters determined? How sensitive is the proposed method to parameter choice?

Minor comments:
-------------------------
1 - In Introduction paragraph 2, there is a typo: laten -> latent
2 - In Sec 2.1, paragraph 3: the algorithm is referred as "above algorithm" but it appears "below" in the paper.
3 - In Sec 2.2, paragraph 3, there is a typo: state-of-art -> state-of-the-art
4 - In Sec 2.2, paragraph 3, there is a typo: laten -> latent

Overall:
-----------
I found the idea of attacking existing OOD detection methods interesting. However, I believe, this paper needs improvement in terms of clarity of the presentation, experimental evaluations and the presentation of the contributions. Therefore, my initial rating is reject for this paper.

---

> ### Author Response · Authors · 2020-11-17
> **Reply to AnonReviewer3**
>
> We thank the reviewer for thorough reading and reviewing our manuscript. Next, we answer the specific questions from the reviewer.
>
> (1) "From the abstract, I infer that the paper has roughly 3 contributions: 1) the paper shows existing OOD detection methods are practically breakable, 2) Glow likelihood-based OOD detection is ineffective, and 3) present a simple theoretical solution with guaranteed performance for OOD detection. However, the 3rd contribution is never mentioned/introduced in the paper until Appendix A where they briefly presents 2 different implementations of an OOD detection method idea. I would not consider this as a contribution since there is no experimental evaluation showing the OOD detection performance of the idea. Also, since this contribution is mentioned in the abstract, I would expect seeing its description and the results in the main paper rather than the Appendix".
>
> Reply:  In the new experiments, we have shown that eight OOD detection methods are breakable under the OOD Attack.  Please see New Discussion (8) for the theoretical solution.
>
> (2) "Until the end of the Introduction, it is not very clear that the main contribution of the paper is a method that attacks OOD detection methods. I liked the motivating example in Figure 1, but the contribution can be given in a more clear way."
>
> Reply:  We will explicitly state the contribution in the revised manuscript.
>
> (3) "In the last paragraph of page 2, it is mentioned that the clip operator can ensure x_out to be OOD after a small modification to x'_out. I found this statement quite vague. How this operation "ensures" that x_out to be OOD after the modification?"
>
> Reply:  we thank the reviewer for pointing out this issue. "ensure" is not the right word, and we will change the description to "the clip operator will limit the difference between x'_out and x_out so that x_out may be OOD".
>
> (4) "In the last paragraph of Section 2.1, it is mentioned that adding initial random noise helps to avoid local minimum caused by a bad initialization. Is this something that you observed empirically? The contribution of the noise is not very clear to me and I think showing results with and without the noise would be very useful to demonstrate how the noise helps to avoid local minima. Also, it would be interesting to show how this loss evolves during the optimization."
>
> Reply:  adding initial random noise is the standard procedure in the PGD optimization technique, and it is theoretically useful. Since in most of our experiments, we use random noise images as the initial samples, this step could be skipped. We will add a figure of loss curves in the Appendix of the revised manuscript, which is basically a curve going down and becoming flat.
>
> (5) "In Sec 2.2 - Each pixel in an 8-bit image can take 256 different values. So, for an image with size 224x224x3, there are 256^(224x224x3) possible images that can be generated; not 8^(224x224x3)."
>
> Reply:  we thank the reviewer for pointing out this bug.
>
> (6) "I didn't quite understand the message in the last paragraph of Sec. 2.2. Why \Omega_in is split into \Omega_{in_clean} and \Omega_{in_noisy}? How this is used in the proposed method?"
>
> Reply:  this is used to explain the feasibility of adversarial training using noisy images generated by adversarial attacks, NOT used in OOD Attack method. An adversarial attack will add a small amount of noise to the input, and the noisy image is still human-recognizable, e.g., a noisy image of a panda still is an image of a panda. The noisy image of the panda is in \Omega_{in_noisy} and the clean image of the panda is in \Omega_{in_clean}.

---

> > ### Author Response · Authors · 2020-11-17
> > **continue...**
> >
> > (7)  "I think that experimental evaluations are not comprehensive enough, 7.1. It is mentioned that implementing OOD detection methods are not necessary since Alg.1 already produces z_out ~= z_in. I don't agree with this since most of the SoTA methods do not rely on only a single representation but multiple representations at different layers [ref1, ref2, ref3]. Therefore, it is crucial to show the performance loss of these methods due to the proposed method."
> >
> > Reply:  we have done new experiments, which includes the evaluation of the method in [ref2] Lee et al., A Simple Unified Framework for Detecting Out-of-Distribution Samples and Adversarial Attacks. (named Mahalanobis)
> >
> > [ref3] "Sastry et al., Detecting Out-of-Distribution Examples with Gram Matrices" presents an OOD detection method using the gram matrices from almost all of the layers of a network. We have tried to use Total Deviation as the target f(x) in OOD attack; however, due to the p-th order gram matrices (p is from 1 to 10), gradients will become inf or nan during backpropagation in the OOD attack algorithm. Therefore, to break this detection method, a gradient free optimization algorithm is needed, which warrants a future study.
> >
> > We cannot find the source code of [ref1] Erdil et al., Unsupervised out-of-distribution detection using kernel density estimation, and some implementation details are missing in that paper. We will cite this paper in the revised manuscript, and we will evaluate this method using our algorithm when the authors release their source code.
> >
> > (8) "Benchmark datasets that have been used in OOD detection paper are usually different than the ones used in the paper. Please see the references above. I wonder why the datasets choice are different than the standard benchmarks?"
> >
> > Reply: in the new experiments, we used the standard datasets, e.g., ImageNet (subset), CIFAR, and SVHN. In the appendix, non-standard datasets such as medical images are used for evaluation. Some DNN systems for medical image diagnosis have been FDA-cleared and are supposed to be able to work independently. We wanted to point out the safety issues related to OOD.
> >
> > (9) "As mention in paragraph 3 of Sec. 4., there are other generative models that show promising results. However, I didn't quite understand comparison with these methods are "out of the scope" while Glow is relevant."
> >
> > Reply:  we indeed evaluated two generative models, Glow and JEM. We will delete the statement of "out of the scope".
> >
> > (10) "The parameters used in different experiments are different than each other. How are these parameters determined? How sensitive is the proposed method to parameter choice?"
> >
> > Reply: for each experiment with an OOD detection method on a dataset, we manually tune the parameters on a few images and use them for the whole dataset. As the experiments are very time-consuming (we have been debugging and running experiments for months), our priority is to make sure the OOD attack works. Based on our observations from parameter tuning, a large number of iterations (N) may lead to a stronger attack (i.e. a lower loss). Using learning rate scheduling for the Adamax optimizer may lead to a stronger attack. We defer a full and quantitative sensitive analysis to our future work.
> >
> > (11) "Minor comments…"
> >
> > Reply: we thank the reviewer for the comments, and we will revise the manuscript accordingly.

---

> > > ### Comment · AnonReviewer3 · 2020-11-21
> > > **Authors' responses are quite sufficient, but incorporating them in the final manuscript requires another review**
> > >
> > > I would like to thank to authors for the detailed rebuttal that addresses the reviewers' concerns. I really appreciate the effort that they put in this period. I believe the new experiments and discussions in the rebuttal would improve the paper's quality remarkably. However, incorporating all these changes and the additional experiments would require another round of review. Therefore, I would like to keep my initial rating for this paper and suggest rejections for ICLR 2021. Having said that, I strongly suggest authors to revise the manuscript by incorporating the new discussion and experiments, and try an another venue.
> > >
> > > I would also like to add more suggestions which might be useful to improve the manuscript:
> > > 1 - As I mentioned above, the additional analysis with the existing methods are quite useful. However, in the current form presented in the rebuttal, they are not quite sufficient. There should be more details about the OOD detection methods used in the experiments. For example, Mahalanobis operates in multiple layers. Therefore, the proposed OOD attack algorithm should modify the input such that the representations in all layers are similar to an in-distribution image. This is a more difficult optimisation problem than updating the input based on a representation at a single layer. Such details requires more discussions and experimental validations to demonstrate the capability of the proposed OOD attack algorithm.
> > >
> > > 2 - As also mentioned by some reviewers, the contribution might seem low since the proposed OOD attack method is quite similar to adversarial attack methods in terms of the optimisation procedures. While the adversarial attack methods updates the input images to maximize the probability for a certain class, the proposed approach does a similar thing the make an internal representation similar to an in-distribution image. This is a very intuitive extension of the adversarial perturbation methods. Although this is interesting, it would be even more interesting to propose a method to tackle with such attacks such as the ones suggested but not validated in the paper. I think proposing such a defense mechanism would improve the contribution.
> > >
> > > 3 - Writing of the paper definitely needs improvement both in terms of language and organisation. Motivation, contribution, method and the experiments should be explained more clearly and in an organised way.

---

> > > > ### Author Response · Authors · 2020-11-21
> > > > **reply: the deadline for discussion is Nov. 24**
> > > >
> > > > (1) " I would like to thank to authors for the detailed rebuttal that addresses the reviewers' concerns. I really appreciate the effort that they put in this period. I believe the new experiments and discussions in the rebuttal would improve the paper's quality remarkably."
> > > >
> > > > Reply: we thank the reviewer for this comment.
> > > >
> > > > (2) "However, incorporating all these changes and the additional experiments would require another round of review. Therefore, I would like to keep my initial rating for this paper and suggest rejections for ICLR 2021. Having said that, I strongly suggest authors to revise the manuscript by incorporating the new discussion and experiments, and try an another venue."
> > > >
> > > > Reply: the deadline for discussion is Nov. 24. the "New Experiments" have provided sufficient details, and we only need to add the text into the paper, which can be done in this weekend.
> > > >
> > > > (3) "I would also like to add more suggestions which might be useful to improve the manuscript: 1 - As I mentioned above, the additional analysis with the existing methods are quite useful. However, in the current form presented in the rebuttal, they are not quite sufficient. There should be more details about the OOD detection methods used in the experiments. "
> > > >
> > > > Reply:  for the details of the OOD detection methods, we refer the reader to the cited papers and their source code. We used the source code of the methods as reference for our experiments, except baseline and deep-ensemble methods, which do not have source code available online but are very easy to understand and implement.
> > > >
> > > > (4) " For example, Mahalanobis operates in multiple layers. Therefore, the proposed OOD attack algorithm should modify the input such that the representations in all layers are similar to an in-distribution image. This is a more difficult optimisation problem than updating the input based on a representation at a single layer. Such details requires more discussions and experimental validations to demonstrate the capability of the proposed OOD attack algorithm."
> > > >
> > > > Reply:  " the proposed OOD attack algorithm should modify the input such that the representations in all layers are similar to an in-distribution image" is simply impossible because it is dimensionality increasing, NOT dimensionality reduction.  We have stated clearly in our original paper that "dimensionality reduction in an encoder provides the opportunity for the existence of the mapping of OOD and in-distribution samples to the same locations in the latent space", and our OOD Attack algorithm is designed to take advantage of this opportunity.
> > > >
> > > > (5) "As also mentioned by some reviewers, the contribution might seem low since the proposed OOD attack method is quite similar to adversarial attack methods in terms of the optimisation procedures. While the adversarial attack methods updates the input images to maximize the probability for a certain class, the proposed approach does a similar thing the make an internal representation similar to an in-distribution image. This is a very intuitive extension of the adversarial perturbation methods. Although this is interesting, it would be even more interesting to propose a method to tackle with such attacks such as the ones suggested but not validated in the paper. I think proposing such a defense mechanism would improve the contribution."
> > > >
> > > > Reply:  As the name of our paper indicates, it is an attack paper, not a defense paper.
> > > >
> > > > In the manuscript, we proposed the subspace saturation training method for OOD detection to show that it may be very difficult to do OOD detection. It is a result of logical reasoning: assuming we want to develop a classification-based OOD detection method that aims to find a decision boundary between in-distribution samples and OOD samples, but we do not want to use any OOD samples, then what can we do? If we use GAN to generate samples and Glow to do bijective mapping, then the saturation rate can be measured by the number of training samples divided by the number of samples needed to saturate the subspace. It would be interesting to study the relationship between the OOD detection rate and the saturation rate. We defer the experiment to our future work because it needs a lot of computing power and time. We will move this part to appendix, which will not affect the main contribution of the paper.
> > > >
> > > >
> > > > (6) Writing of the paper definitely needs improvement both in terms of language and organisation. Motivation, contribution, method and the experiments should be explained more clearly and in an organised way.
> > > >
> > > > Reply:  given the 8-page limit, not everything can be done perfectly. From the comments of all the reviewers, I believe the reviewers have no difficulty understanding what we have done in this work.

---

> > > > > ### Comment · AnonReviewer3 · 2020-11-21
> > > > > **New experiments are quite useful, but not sufficient without all the details, discussion, and improvement in organisation**
> > > > >
> > > > > Please find my point to point responses below:
> > > > >
> > > > > (2) I would be happy to review the revised paper and update my score if I believe the paper is ready for publication.
> > > > >
> > > > > (3) - (4) The core success of Mahalanobis comes from OOD detection at multiple layers. So, a proper evaluation of an OOD attack method should be performed in the most optimum setting of the OOD detection method. Therefore, the details about the setting that are used in the experiments are crucial. It doesn't help simply referring authors to the original paper since the original papers don't state that how you performed your experiments.
> > > > >
> > > > > (5) Authors state in the abstract that "At last, we present a simple theoretical solution with guaranteed performance for OOD detection". Since this is claimed by the authors as one of the contributions, this means that the paper is NOT only an attack paper but also a defense paper. I believe, a contribution claimed in the abstract should be validated to some extend in the main paper whereas appendix can contain additional validations. Without a validated claim, I cannot consider this as a contribution. Moreover, this claim is not validated in the appendix either.
> > > > >
> > > > > * Response to the final comment regarding writing:
> > > > > - All reviewers might have understood the paper; but this does not mean that the paper doesn't have any problem regarding clarity and organisation. As indicated by ALL reviewers, clarity and organisation of the paper requires improvement.

---

### Official Review · AnonReviewer4 · 2020-10-25

**Rating:** 4
**Confidence:** 4

**Review:**

Summary of paper: this work shows that adversarial perturbations can make any OOD image, map to the same latent code as an in-distribution - creating an attack on confidence-based or flow-based ODD detection methods. Results are shown on a few datasets with some attempts at evaluation.

Novelty: Unfortunately, I don't think there is much new here. Adversarial attacks are of course well known - I am not sure that attacks on intermediate latent codes present novelty either. Adversarial attacks against anomaly detection methods have also been investigated before (e.g. [1] [2], although their setting is a little different) and there is nothing in the proposed method that is particularly tailored to OOD.

Evaluation: the evaluation is not extensive - only one adversarial attack is investigated and no reasonable baselines have been selected. I am not sure that MAPE is an appropriate metric - it really depends on the allowed perturbation. Ss the allowed perturbation small enough? do the perturbed images look realistic - if I understood Fig.12, they don't - but the caption there is not clear.

Clarity - the paper is not particularly clearly written - although the idea is simple enough. E.g. I don't see the scatter plots clearly explained, the analysis in Sec. 2.2 is very dense for a fairly simple idea.

Overall: ultimately, this is conceptually repeating the same thing as any other adversarial examples work, perturbations can make a network confident that any image has the label of another image - and this obviously would overcome confidence based OOD detection methods. I therefore do not see a strong contribution by this work. As there is also very limited methodological novelty, I do not think it should be accepted.

###############################################################################

I understand the distinction the authors are trying to draw between adversarial examples for anomaly detection and fooling OOD to think that images are in distribution where in fact they are OOD. I still don't think that technically or conceptually, there is much difference. The authors presented many fresh results during the rebuttal (which might have been better presented just as a table in the manuscript, rather than on this thread). The experiments can form a part of a resubmission of this work, that will incorporate the extensive comments presented by the current reviews.

[1] Rigaki, Maria. "Adversarial deep learning against intrusion detection classifiers." (2017).
[2] Bergman and Hoshen, Classification-based anomaly detection for general data, ICLR'20

---

> ### Author Response · Authors · 2020-11-17
> **Reply to AnonReviewer4**
>
> We thank the reviewer for thorough reading and reviewing our manuscript.  Here, we answer the specific questions from the reviewer.
>
> (1)"Unfortunately, I don't think there is much new here. Adversarial attacks are of course well known - I am not sure that attacks on intermediate latent codes present novelty either. Adversarial attacks against anomaly detection methods have also been investigated before (e.g. [1] [2], although their setting is a little different) and there is nothing in the proposed method that is particularly tailored to OOD."
>
> Reply: we presented an approach to generate OOD samples and evaluate OOD detection methods, which is not done in the eight OOD detection papers published at ICLR and NeurIPS. We note that adversarial attack and OOD attack are doing completely different things to neural networks.
>
> [1] Rigaki, Maria. "Adversarial deep learning against intrusion detection classifiers." (2017). In this paper, FSGM and JSMA methods are used to generate adversarial samples. This is a study of adversarial robustness, NOT OOD Attack issue presented in our manuscript, NOT the OOD detection issue defined in the baseline method (ICLR 2017) in https://arxiv.org/abs/1610.02136 and investigated in the other seven papers at ICLR and NeurIPS.
>
> [2] Bergman and Hoshen, Classification-based anomaly detection for general data, ICLR'20. For image related applications, this paper investigated a special case of OOD detection, also known as one class classification, which is very different from the OOD detection tasks on dataset level: e.g. MNIST to be in-distribution and Omniglot to be OOD as demonstrated in the paper of the baseline method. Our study focused on OOD detection issues on the dataset level.
>
> (2) "Evaluation: the evaluation is not extensive - only one adversarial attack is investigated and no reasonable baselines have been selected. I am not sure that MAPE is an appropriate metric - it really depends on the allowed perturbation. Ss the allowed perturbation small enough? do the perturbed images look realistic - if I understood Fig.12, they don't - but the caption there is not clear."
>
> Reply: adversarial attack and OOD attack are doing completely different things. For an adversarial attack, the perturbation must not be too large, and the perturbed images should be human-recognizable. For an OOD attack, the generated OOD samples can be arbitrary, such as random noises or wired images. By the definition of OOD, an OOD sample does not and should not look similar to the in-distribution samples.
>
> (3) "Clarity - the paper is not particularly clearly written - although the idea is simple enough. E.g. I don't see the scatter plots clearly explained"
>
> Reply: As explained by the figure captions, the scatter plots show the feature vectors (z).
>
> (4) "the analysis in Sec. 2.2 is very dense for a fairly simple idea. "
>
> Reply: OOD is a complex issue, and it should be thoroughly analyzed.
>
> (5) "Overall: ultimately, this is conceptually repeating the same thing as any other adversarial examples work, perturbations can make a network confident that any image has the label of another image - and this obviously would overcome confidence based OOD detection methods. I therefore do not see a strong contribution by this work. As there is also very limited methodological novelty, I do not think it should be accepted."
>
> Reply: Adversarial attack and OOD attack are doing completely different things to neural networks. Please see our new experiments and discussion. We note that we have evaluated eight dataset-level OOD detection methods, and we believe our work has made a considerable contribution to the field.

---

### Official Review · AnonReviewer2 · 2020-10-27

**Rating:** 3
**Confidence:** 5

**Review:**

Summary:

The paper defines an out-of-distribution attack, a process which drives an out-of-distribution (OOD) input to have the same latent representation to an inlier. The paper also analyzes that an encoder is inevitably vulnerable to out-of-distribution attack when its latent dimensionality is smaller than the dimensionality of the input.

Decision:

Reject

Strength:

The paper addresses an important vulnerability of classifier-based OOD detection. As classifier-based method is one of the currently dominating approaches for OOD detection, investigating its weakness is a significant contribution to the research community.

Weakness:

The proposed attack algorithm is significantly similar to the previously known adversarial attack algorithms and therefore seems trivial.

The main quantitative result, Table 1, is not very convincing. I suggest the authors  provide AUC scores computed before and after OOD attack, so that the difference clearly shows that the proposed attack causes a decrease in OOD detection performance.

The paper should benchmark the proposed attack algorithm against state-of-the-art OOD detection methods. Currently, only a relatively simple method of Hendrycks and Gimpel, 2016 is used. The method should include at least [1,2,3] to show the effectiveness of the proposed attack. At least some of these OOD detection methods may be able to resist the proposed attack. For example, multiple hidden layer representations from a classifier are used in [1], and therefore it can still detect OOD even if a specific latent representation is under attack.

The organization of the paper needs to be improved. In the last sentence of the abstract, "a simple theoretical solution" is mentioned but is only addressed in Appendix. If it is a contribution that is important enough to be mentioned in the abstract, it should be covered in depth in the main manuscript instead of Appendix.

Minor comments:
- The visibility of figures are poor. The axis titles and MAPE values in Figure 2, 3, 4, 5 should be larger.
- In Section 2.2, 8^{224*224*3} should 256^{224*224*3}. An 8-bit integer can represent 256 values.
- The captions of Figure 3 and Figure 4 are the same.
- The captions of Figure 5 and Figure 6 are too close.

Typos:
laten → latent (Section 1 paragraph 2 line 10)
Dicussion → Discussion (Section 4 title)
Difficulty → difficult (Section 4 paragraph 4 first line)

[1] Lee, Kimin, et al. "A simple unified framework for detecting out-of-distribution samples and adversarial attacks." Advances in Neural Information Processing Systems. 2018.
[2] Liang, Shiyu, Yixuan Li, and Rayadurgam Srikant. "Enhancing the reliability of out-of-distribution image detection in neural networks." arXiv preprint arXiv:1706.02690 (2017).
[3] Grathwohl, Will, et al. "Your classifier is secretly an energy based model and you should treat it like one." arXiv preprint arXiv:1912.03263 (2019).

---

> ### Author Response · Authors · 2020-11-17
> **Reply to AnonReviewer2**
>
> First, we thank the reviewer for thorough reading and reviewing our manuscript. Next, we answer the specific questions from the reviewer.
>
> (1) "The proposed attack algorithm is significantly similar to the previously known adversarial attack algorithms and therefore seems trivial."
>
> Reply: about the novelty, please read New Discussion (6) and (7). We note that adversarial attack and OOD attack are doing completely different things to neural networks. Please read the Clarification.
>
> (2) "The main quantitative result, Table 1, is not very convincing. I suggest the authors provide AUC scores computed before and after OOD attack, so that the difference clearly shows that the proposed attack causes a decrease in OOD detection performance."
>
> Reply: we evaluated eight OOD detection methods in the new experiments. What are "AUC scores computed before OOD attack"? the AUC scores in the papers of the OOD detection methods?
>
> (3) "The paper should benchmark the proposed attack algorithm against state-of-the-art OOD detection methods. Currently, only a relatively simple method of Hendrycks and Gimpel, 2016 is used. The method should include at least [1,2,3] to show the effectiveness of the proposed attack. At least some of these OOD detection methods may be able to resist the proposed attack. For example, multiple hidden layer representations from a classifier are used in [1], and therefore it can still detect OOD even if a specific latent representation is under attack."
>
> Reply: we evaluated the three OOD detection methods in the new experiments.
>
> (4) "The organization of the paper needs to be improved. In the last sentence of the abstract, "a simple theoretical solution" is mentioned but is only addressed in Appendix. If it is a contribution that is important enough to be mentioned in the abstract, it should be covered in depth in the main manuscript instead of Appendix."
>
> Reply: please read New Discussion (8)
>
> (5) "Minor comments…"
>
> Reply: we thank the reviewer for the comments, and we will revise the manuscript accordingly. The figures are in high-resolution; please zoom in on the computer screen.

---

### Official Review · AnonReviewer1 · 2020-10-28
**Review AnonReviewer1**

**Rating:** 4
**Confidence:** 4

**Review:**


**UPDATE**

I acknowledge that I have read the author responses as well as the other reviews. Overall, I appreciate the clarifications and added experiments given by the authors.

My concerns about the low novelty of the presented algorithm and findings remain, however, as I find the OOD attack to be only a slight modification of existing adversarial attacks.

I also appreciate that the defense solution claim has been weakened and moved to the appendix, yet these promises are still left to be validated.

Lastly, I find all the many added experiments positive, but these have significantly changed the content of the initial submission at this point, which is somewhat out of scope of the ICLR rebuttal phase (see Q4 in the FAQ of the Reviewer Guide: https://iclr.cc/Conferences/2021/ReviewerGuide).

For these reasons, I would keep my recommendation to reject this work for ICLR 2021, but I encourage the authors to further improve and re-submit the now extended work to some future venue.

#####


**Summary**

This paper presents an algorithm for out-of-distribution (OOD) attacks on neural networks. Given a target network and an initial OOD input, the proposed algorithm performs projected gradient descent (PGD) w.r.t. the input to obtain an output that is close to the embedding of some desired in-distribution sample. The paper makes the case that dimensionality reduction is one key reason that makes standard deep networks vulnerable to OOD attacks, due to the non-bijective nature of such mappings. An experimental evaluation on two ImageNet- pretrained classifiers (ResNet-18 and DenseNet-121) is presented, where a subset of ImageNet serves as in-distribution and chest x-ray, lung-CT, and noise images serve as OOD samples, that demonstrates empirically that these networks can be attacked with the proposed algorithm. Another experiment on the likelihood-based normalizing flow model Glow is carried out which demonstrates that such bijective, dimensionality-preserving deep generative models are also breakable by the proposed attack. Finally, a theoretical sketch for a solution of the problem is described.


**Pros**
+ The paper presents a simple OOD attack algorithm and demonstrates empirically that OOD samples can be perturbed such that they map to the embedding of some arbitrary in-distribution example.
+ The paper makes a plausible argument that dimensionality reduction enables OOD attacks. Inversely, it is argued that reconstruction methods, which map back to the original space, are favorable for OOD detection for which the proposed attack is also ineffective.
+ Attacking the dimensionality-preserving, likelihood-based Glow model is an interesting experiment to consider in the context of the presented dimensionality reduction argument.
+ The paper is structured well and overall well-placed into existing literature.

**Cons**
- I find the novelty of the presented algorithm and experimental findings to be rather low.
- The experimental evaluation of the proposed attack is limited to standard classifiers and does not include deep networks that have been trained to increase OOD robustness [1, 2, 4, 3].
- The defense solution is only a sketch and makes promises that are left to be validated.
- There are many errors in language and grammar, e.g. wrong use of tense, missing articles, etc. (see minor comments below)


**Recommendation**

I think the current paper is ok but not good enough (score: 4) due to (i) low novelty, (ii) a limited experimental evaluation, and (iii) solution claims that are left to be validated.

(i) Though I see and agree that the OOD attack setting is slightly different to adversarial attacks, I find the brittleness of standard classifiers in this regard not surprising. In particular, the OOD attack has a greater degree of freedom since any arbitrary OOD input can be used as a starting point for perturbation (pure noise is also OOD, as remarked in the paper), i.e. there is no similarity constraint on the input as there is for adversarial attacks. Moreover, I find the algorithmic novelty to be low as well, since the proposed algorithm essentially is a slight adaptation of previously introduced projected gradient descent (PGD) attacks.

(ii) Following (i), I think the current experimental evaluation is limited and the findings are not surprising for the two standard, pre-trained classifiers (ResNet-18 and DenseNet-121). There exist many approaches that have shown to improve OOD robustness [1, 2, 4, 3], which should be included in the analysis. It would be interesting to see how these approaches perform and compare, which could be insightful for improving OOD robustness.

(iii) Proposing an attack begs the question what possible defenses could be. Currently, the main paper is only phenomenological, i.e. demonstrates that OOD attacks are an open issue, but the description of a possible defense at the end of the paper and in the appendix makes only a solution claim which is left to be validated. I don’t say or think such a solution would be necessary for an interesting and valid contribution, as OOD detection poses a hard problem which likely lacks a simple solution, but the current solution is a mere sketch making promises with questions left open. For example, how can a sufficient space saturation be achieved with a finite sample in practice? Which measure to use?


**Additional feedback and ideas for improvement**

I think the OOD detection problem, both from an attack and defense perspective is relevant and of great interest to the community, which is why I encourage the authors to build upon and extend the current manuscript. Some ideas:

- Including methods that have shown to improve OOD robustness [1, 2, 4, 3] would greatly improve the value of the analysis. This could be insightful to further improve OOD robustness and see pros/cons of existing approaches.
- Does adversarial training, as mentioned in Section 2.2, also help OOD robustness?
- Implementing and validating the presented OOD detection solution would be interesting.
- Further investigate the reason why the dimensionality-preserving Glow model can be attacked as well, as this suggests dimensionality reduction is only a part of the issue.
- ‘Could we design an evaluation method (experimental or analytical) that does not rely on OOD samples?’ I think this is a worthwhile question to ask and research.
- Add missing related work [5].


**Minor Comments**

1. ‘The algorithm needs a weak assumption that $f(x)$ is differentiable’ I think sub-differentiable would be sufficient, correct?
2. ‘If the neural network only uses ReLU activation, then the input-output relationship can be exactly expressed as a linear mapping’ Only piecewise linear mapping, right?
3. ‘[...], the above mentioned classification-based OOD detection is theoretically almost ineffective [...]’; ‘[...], then it is highly possible that the entire latent space is crawling with the shadows of OOD samples.’; ‘[...], there are a huge number of “holes” in the space, [...]’ Rather avoid such vague formulations.
4. Published works should be included as such in the references, not by their preprint reference (e.g., [2] appeared at ICLR).
5. A list of citations in a sentence should be concatenated within single parentheses/brackets, separated by commas (not individual parentheses/brackets per citation).
6. When referring to a specific section, section should be capitalized, e.g. Section 3.
7. Abstract: ‘Deep Neural *Networks (DNNs)*, especially convolutional neural *networks (CNNs)*, *have* become ...’ It is a class of models, hence plural.
8. Algorithm 1: ‘[...] not similar to *any sample* in the dataset.’; ‘$\alpha$ the learning rate of *the* optimizer’; $h$ used in the Algorithm is not defined; I would use $\nabla_x J$ instead of $J'$ for the gradient to stress the derivative is w.r.t. the input.
9. Many figure labels are tiny and hard to read.
10. The captions of Figure 3 + 4 are (almost) identical. Add a label indicating the network.
11. Section 1: ‘It was shown by *Nguyen et al. (2015)* [...], and *an* evolutionary algorithm was used [...]’
12. Section 1: ‘Since then, many methods *have been* proposed for OOD detection [...]’
13. Section 1: ‘For instance, *Hendrycks et al. (2016) show* that a classifier’s prediction *probabilities* of OOD examples tend to be *more uniform* [...]’ ‘lower’ is ambiguous as the softmax probabilites always add up to 1, right?
14. Section 1: ‘For *the evaluation of OOD detection methods*, an OOD detector is *usually* trained [...]’
15. Section 1: ‘[...] pre-trained on *the* ImageNet dataset.’
16. Section 1: ‘[...] which could be any kind of *image* (even random *noise*) [...]’.
17. Section 1: ‘[...], which is the input to the last fully-connected linear layer before *the* softmax operation.’
18. Section 1: ‘[...], and *one* fix to the problem could be using likelihood ratio [...]’
19. Section 1: ‘[...], we will show that *the* OOD sample’s likelihood score from the Glow model [...]’
20. Section 2: ‘[...] projected gradient descent (PGD) which is used for adversarial *attacks*.’
21. Section 2: ‘In practice, *Algorithm 1 can be repeated* many times [...]’
22. Section 2: ‘[...], the algorithm will have a better chance to avoid *a local minimum* [...]’
23. Section 2: ‘This *is* simply because the vectors in a lower-dimensional space [...]’
24. Section 2: ‘[...], which is the Pigeonhole Principle.’ Citation?
25. Section 2: ‘Usually, *the* training set is only a subset’
26. ‘3 EXPERIMENT’ Do not capitalize and ‘3 Experimental Evaluation’?
27. Section 3: ‘*A* Nvidia Titan V GPU was used [...]’
28. Section 4: ‘We would like to point out that it is *difficult* to evaluate [...]’


#####

**References**

[1] B. Lakshminarayanan, A. Pritzel, and C. Blundell. Simple and scalable predictive uncertainty estimation using deep ensembles. In NIPS, pages 6402–6413, 2017.

[2] K. Lee, H. Lee, K. Lee, and J. Shin. Training confidence-calibrated classifiers for detecting out-of-distribution samples. In ICLR, 2018.

[3] K. Lee, K. Lee, H. Lee, and J. Shin. A simple unified framework for detecting out-of-distribution samples and adversarial attacks. In NeurIPS, pages 7167–7177, 2018.

[4] S. Liang, Y. Li, and R. Srikant. Enhancing the reliability of out-of-distribution image detection in neural networks. In ICLR, 2018.

[5] A. Meinke and M. Hein. Towards neural networks that provably know when they don’t know. In ICLR, 2020.

---

> ### Author Response · Authors · 2020-11-17
> **reply to AnonReviewer1**
>
> First, we thank the reviewer for thorough reading and reviewing our manuscript. Next, we answer the specific questions from the reviewer.
>
> (1) "I find the novelty of the presented algorithm and experimental findings to be rather low."
>
> Reply: we have provided new experimental findings. About the novelty, please read New Discussion (6) and (7)
>
> (2) "The experimental evaluation of the proposed attack is limited to standard classifiers and does not include deep networks that have been trained to increase OOD robustness [1, 2, 4, 3]."
>
> Reply: we evaluated the four methods in the new experiments
>
> (3) "The defense solution is only a sketch and makes promises that are left to be validated."
>
> Reply: Please read New Discussion (8)
>
> (4) "Though I see and agree that the OOD attack setting is slightly different to adversarial attacks, I find the brittleness of standard classifiers in this regard not surprising. In particular, the OOD attack has a greater degree of freedom since any arbitrary OOD input can be used as a starting point for perturbation (pure noise is also OOD, as remarked in the paper), i.e. there is no similarity constraint on the input as there is for adversarial attacks."
>
> Reply: our new experiments show that eight classification-based OOD detection methods do not work well under the OOD attack, which is new information.
>
> (5) "Moreover, I find the algorithmic novelty to be low as well, since the proposed algorithm essentially is a slight adaptation of previously introduced projected gradient descent (PGD) attacks."
>
> Reply: about the novelty, please read New Discussion (6) and (7).
>
> (6) "I think the current experimental evaluation is limited and the findings are not surprising for the two standard, pre-trained classifiers (ResNet-18 and DenseNet-121). There exist many approaches that have shown to improve OOD robustness [1, 2, 4, 3], which should be included in the analysis. It would be interesting to see how these approaches perform and compare, which could be insightful for improving OOD robustness."
>
> Reply: we evaluated the four methods in the new experiments.
>
> (7) "Proposing an attack begs the question what possible defenses could be. Currently, the main paper is only phenomenological, i.e. demonstrates that OOD attacks are an open issue, but the description of a possible defense at the end of the paper and in the appendix makes only a solution claim which is left to be validated. I don’t say or think such a solution would be necessary for an interesting and valid contribution, as OOD detection poses a hard problem which likely lacks a simple solution, but the current solution is a mere sketch making promises with questions left open. For example, how can a sufficient space saturation be achieved with a finite sample in practice? Which measure to use?"
>
> Reply: please read New Discussion (8)
>
> (8) "I think the OOD detection problem, both from an attack and defense perspective is relevant and of great interest to the community, which is why I encourage the authors to build upon and extend the current manuscript. Some ideas:  Including methods that have shown to improve OOD robustness [1, 2, 4, 3] would greatly improve the value of the analysis. This could be insightful to further improve OOD robustness and see pros/cons of existing approaches. Does adversarial training, as mentioned in Section 2.2, also help OOD robustness?"
>
> Reply: we have evaluated the four methods in new experiments. The Deep Ensembles (NeurIPS 2017, https://arxiv.org/abs/1612.01474) used adversarial training, and the evaluation results show that it did not work. This is not surprising because noisy images with adversarial noises are still in-distribution (e.g. a noisy image of a panda still is an image of a panda). Adversarial samples and OOD samples are completely different.
>
> (9) "Implementing and validating the presented OOD detection solution would be interesting."
>
> Reply: please read New Discussion (8)
>
> (10) Further investigate the reason why the dimensionality-preserving Glow model can be attacked as well, as this suggests dimensionality reduction is only a part of the issue."
>
> Reply: please read New Discussion (2)
>
> (11) "Could we design an evaluation method (experimental or analytical) that does not rely on OOD samples?’ I think this is a worthwhile question to ask and research."
>
> Reply: we thank the reviewer for this comment.
>
> (12) "Add missing related work [5]"
>
> Reply: we will add it to the revised manuscript.
>
> (13) "Minor Comments…"
>
> Reply: we thank the reviewer for the comments, and we will incorporate those into the revised manuscript.

---

### Author Response · Authors · 2020-11-17
**New Discussion**

(1) The nine OOD detection methods do not work well in our experiments.

(2) It is known that Glow NLL score cannot be used to distinguish between OOD and in-distribution samples. A recent study proposed a fix by combining NLL with input complexity. In our new experiment, we show that this method is ineffective as well. We hypothesize that the main cause of its vulnerability to OOD attack is that there is not enough training data in high dimensional space. Glow is a bijective mapping between input x and output z. Since the size of the training set is tiny compared to the whole input space, Glow is free to map OOD samples to any places in the latent space, and as a result, OOD samples can be assigned with a large range of probability scores.

(3) For the JEM method [7], AUROC using ||d_logp(x)/d_x|| is quite high. This is due to the fact that our algorithm does not directly attack on the gradient image d_logp(x)/d_x, which has the same shape as the input x. As shown by Table 3 in the JEM paper, "for tractable likelihood methods we find this predictor is anti-correlated with the model’s likelihood and neither is reliable for OOD detection". Further investigation is needed to determine whether it is a general OOD metric.

(4) Energy-based models (EBMs), such as JEM, can generate OOD samples during training, which may explain why the OOD attack failed when the initial OOD sample was random noise. If we take a closer look at the sampling procedure (e.g. Langevin dynamics) and the objective function, it is easy to find out EBM training algorithm is trying to pull down the energy scores of positive (in-distribution) samples and pull up the energy scores of negative (OOD) samples (see https://arxiv.org/abs/1903.08689), which is similar to the basic idea of adversarial training.

(5) We visually inspected the generated OOD images and have some interesting findings. Using random noise images as the initial OOD samples, the generated OOD images for the classification-based methods look like random noises, except the JEM, which is an energy-based model. Using the color spiral as the initial OOD sample, many of the generated OOD images for the JEM are really weird: they are like the images of Frankenstein's monsters: randomly put some parts of objects together, twist/deform them, and then pour some paint on them. It may be difficult for neural networks to learn what is an object (e.g. airplane) just from images and class labels.

(6) From the perspective of optimization, iterative projected gradient descent (PGD) optimization has a long history and can be traced back to 1945: "Abraham Wald. Statistical decision functions which minimize the maximum risk. In Annals of Mathematics, 1945", which is cited in the PGD-based adversarial attack paper:  https://arxiv.org/abs/1706.06083 (ICLR 2018). However, none of the eight OOD detection papers used PGD-style algorithms to generate OOD samples for method evaluation. In the manuscript, we did not state we invented PGD optimization technique, and we adapted it for OOD sample generation, which is the OOD Attack Algorithm 1.

(7) The research on adversarial robustness began as early as the work of Goodfellow et al. in https://arxiv.org/abs/1412.6572 (ICLR 2015). Adversarial robustness issue refers to the fact that adding a small amount of noise to the input may cause a significant change in the output of a DNN model. Adversarial attack and OOD attack are doing completely different things to neural networks, and please read our Clarification. We are surprised that PGD-style OOD attack algorithms were not used for method evaluation in these OOD detection papers published at ICLR and NeurIPS for such a long time until our work.

(8) In the manuscript, we proposed the subspace saturation training method for OOD detection. It is a result of logical reasoning: assuming we want to develop a classification-based OOD detection method that aims to find a decision boundary between in-distribution samples and OOD samples, but we do not want to use any OOD samples, then what can we do? If we use GAN to generate samples and Glow to do bijective mapping, then the saturation rate can be measured by the number of training samples divided by the number of samples needed to saturate the subspace. It would be interesting to study the relationship between the OOD detection rate and the saturation rate. We defer the experiment to our future work because it needs a lot of computing power and time. We will move this part to appendix, which will not affect the main contribution of the paper.

---

> ### Author Response · Authors · 2020-11-23
> **update**
>
> the statement "For the JEM method [7], AUROC using ||d_logp(x)/d_x|| is quite high" is no longer valid.
>
> -||d_logp(x)/d_x||  is used as the OOD score in the JEM method , and AUROC  is less than 0.5
>
> Our algorithm was unable to break a recent method (Meinke & Hein, 2020, Towards neural networks that provably know when they don’t know), mostly because this method utilizes Gaussian mixture models (GMMs) in the input space. However, GMMs have convergence issues for high dimensional data. Thus, it would be difficult to use the method for medical image related applications. (medical images are very large)

---

### Author Response · Authors · 2020-11-17
**New Experiments**

We have used the OOD Attack algorithm to evaluate nine OOD detection methods with papers published at ICLR and NeurIPS. The results are summarized here.

---

> ### Author Response · Authors · 2020-11-17
> **New Experiment 1**
>
> We have used the OOD Attack algorithm to evaluate eight OOD detection methods with papers published at ICLR and NeurIPS. The results are summarized here:
>
> [1] Baseline (ICLR 2017, https://arxiv.org/abs/1610.02136)
>
> Dataset: ImageNet subset that has 1000 images in 200 categories.
> DNNs: Resnet-18 and Densenet-121 pre-trained on ImageNet and imported from torchvision
>
> Parameters for Resnet-18: ε=5, N=1e4, α=ε/100 with X-ray and CT; ε=20, N=1e4, α=ε/100 with random noise.
>
> Parameters for Densnet-121: ε=5, N=1e4, α=ε/100 with X-ray and CT; ε=30, N=1e4, α=ε/100 with random noise.
>
> Target: f(x) is the vector input to the last linear layer.
>
> Results on Resnet-18: AUROC is 0.500 when the initial OOD sample is a CT image; AUROC is 0.500 when the initial OOD sample is an X-ray image; AUROC is 0.500 when the initial OOD sample is a noise image.
>
> Results on Densenet-121: AUROC is 0.500 when the initial OOD sample is a CT image; AUROC is 0.500 when the initial OOD sample is an X-ray image; AUROC is 0.500 when the initial OOD sample is a noise image.
>
> Note: by using learning rate scheduling for the Adamax optimizer, the AUROC scores are further reduced to 0.500, compared to scores in the table-1 in the original manuscript.

---

> ### Author Response · Authors · 2020-11-17
> **New Experiment 2**
>
> [2] ODIN (ICLR 2018, https://arxiv.org/abs/1706.02690)
>
> Dataset: ImageNet subset that has 1000 images in 200 categories.
>
> DNNs: Resnet-18 and Densenet-121 pre-trained on ImageNet and imported from torchvision
>
> Parameters for Resnet-18: ε=5, N=1e4, α=ε/100 with X-ray and CT; ε=20, N=1e4, α=ε/100 with random noise.
>
> Parameters for Densnet-121: ε=5, N=1e4, α=ε/100 with X-ray and CT; ε=30, N=1e4, α=ε/100 with random noise.
>
> Target: f(x) is the vector input to the last linear layer.
>
> Results on Resnet-18: AUROC is in the range of 0.500 to 0.552 when the initial OOD sample is a CT image; AUROC is in the range of 0.500 to 0.550 when the initial OOD sample is an X-ray image; AUROC is in the range of 0.500 to 0.504 when the initial OOD sample is a random noise image.
>
> Results on Densenet-121: AUROC is in the range of 0.500 to 0.568 when the initial OOD sample is a CT image; AUROC is in the range of 0.500 to 0.570 when the initial OOD sample is an X-ray image; AUROC is in the range of 0.500 to 0.501 when the initial OOD sample is a random noise image.
>
> Note: we tested the method with three temperature values (1, 50, 100) and three perturbation magnitudes (0.001, 0.005, 0.01), i.e., 9 possible combinations, which is why we have a range of AUROC scores.

---

> > ### Author Response · Authors · 2020-11-23
> > **update**
> >
> > We also evaluated the method on CIFAR10. The results are reported in the revised paper. AUROC scores are close to 0.500.

---

> ### Author Response · Authors · 2020-11-17
> **New Experiment 3**
>
> [3] Mahalanobis (NeurIPS 2018, https://arxiv.org/abs/1807.03888)
>
> Datasets: CIFAR10 and CIFAR100
>
> DNNs: Resnet-34 downloaded from https://github.com/pokaxpoka/deep_Mahalanobis_detector/
>
> Parameters: ε=10, α= ε/100, N=1e3
>
> Target-1: f(x) is the concatenation of the feature vectors used to calculate the Mahalanobis distances.
> AUROC is 0.467 on CIFAR10 vs OOD when preprocessing perturbation magnitude is 0;
> AUROC is 0.179 on CIFAR10 vs OOD when preprocessing perturbation magnitude is 0.01;
> AUROC is 0.604 on CIFAR100 vs OOD when preprocessing perturbation magnitude is 0;
> AUROC is 0.377 on CIFAR100 vs OOD when preprocessing perturbation magnitude is 0.01; The results show that preprocessing made things worse.
>
> Target-2: f(x) is the average of the Mahalanobis distances
> AUROC is 0.500 on CIFAR10 vs OOD when preprocessing perturbation magnitude is in {0, 0.01};
> AUROC is 0.500 on CIFAR100 vs OOD when preprocessing perturbation magnitude is in {0, 0.01}.
>
> Note: Since in practice, it is infeasible to have a validation set of OOD samples, the final distance is the average of the Mahalanobis distances from multiple layers.

---

> ### Author Response · Authors · 2020-11-17
> **New Experiment 4**
>
> [4] Outlier Exposure (ICLR 2019, https://arxiv.org/abs/1812.04606)
>
> Datasets: SVHN, CIFAR10 and CIFAR100 (test sets)
>
> DNNs: three models downloaded from https://github.com/hendrycks/outlier-exposure. The models are named "oe_scratch" by the authors.
>
> Parameters:  ε=10, α= ε/100, N=1e4
>
> The initial OOD sample is a random noise image
>
> Target: f(x) is the vector (a.k.a. logits) input to softmax.
>
> AUROC is 0.500 on SVHN vs OOD.
>
> AUROC is 0.500 on CIFAR10 vs OOD.
>
> AUROC is 0.500 on CIFAR100 vs OOD.

---

> ### Author Response · Authors · 2020-11-17
> **New Experiment 5**
>
> [5] Deep Ensembles (NeurIPS 2017, https://arxiv.org/abs/1612.01474)
>
> Dataset: CIFAR10 (test set)
>
> DNNs: 6 models downloaded from
> https://github.com/BorealisAI/mma_training/tree/master/trained_models. The names of the moels are cifar10-L2-MMA-1.0-sd0, cifar10-L2-MMA-2.0-sd0, cifar10-L2-OMMA-1.0-sd0, cifar10-L2-OMMA-2.0-sd0, cifar10-Linf-MMA-12-sd0, cifar10-Linf-OMMA-12-sd0
>
> Classification accuracy of the ensemble on test set is 89.85%
>
> Parameters: ε=10, α= ε/100, N=1e4
>
> The initial OOD sample is a random noise image.
>
> Target: f(x) is the vector (a.k.a. logits) input to softmax.
>
> AUROC is 0.500 on CIFAR10 vs OOD.
>
> Note: the original paper states "we used 5 networks in our ensemble". Thus, an ensemble of 6 models should be enough. We did not find pre-trained models of the paper. Therefore, we used 6 pre-trained models (wide residual networks) from someone's recent work on adversarial robustness, in which the SOTA adversarial training method is named MMA (ICLR 2020, https://arxiv.org/abs/1812.02637). The models were trained to be robust against adversarial noises in a large range.

---

> > ### Author Response · Authors · 2020-11-23
> > **discussion**
> >
> >
> > The results from New Experiment 5  show that networks trained to be robust to adversarial attacks may not be robust to OOD attacks.
> >
> > Compared to adversarial attacks and defenses, it is much more difficult to defend against OOD attacks. Adversarial attacks and OOD attacks are doing completely different things to neural networks, although the attack algorithms may use similar optimization techniques. For image classification applications, an adversarial attack will add a small amount of noise to the input (clean) image, and the resulting noisy image is still human-recognizable. Therefore, the magnitudes of adversarial noises are constrained. For  example, a noisy image of a panda is still an image of the panda. By the judgment of humans, the noisy image and the clean image are the images of the same object, and the two images should be classified into the same class. Compared to adversarial samples, OOD samples, which can be generated by our OOD Attack algorithm, have much more freedom (e.g. they can be random noises), as long as they do not look like in-distribution samples. Thus, OOD detection is very challenging.

---

> ### Author Response · Authors · 2020-11-17
> **New Experiment 6**
>
> [6] Confidence-calibrated classifiers (ICLR 2018, https://arxiv.org/abs/1711.09325)
>
> Dataset: SVHN and CIFAR10 (test sets)
>
> DNNs: two VGG13 models, one trained on SVHN and the other trained on CIFAR10. The source code is provided by the authors at https://github.com/alinlab/Confident_classifier.
>
> Parameters:  ε=10, α= ε/100, N=1e4
>
> The initial OOD sample is a random noise image.
>
> Target: f(x) is the vector input to the "Classifier" module.
>
> AUROC is 0.501 on SVHN vs OOD
>
> AUROC is 0.576 on CIFAR10 vs OOD.

---

> ### Author Response · Authors · 2020-11-17
> **New Experiment 7**
>
> [7] JEM (ICLR 2020, https://arxiv.org/abs/1912.03263)
>
> Dataset: CIFAR10 (test set)
>
> DNN: a wide residual network downloaded from https://github.com/wgrathwohl/JEM
>
> Parameters:  ε=20, α= ε/100, N=1e4
>
> The initial OOD sample is a color spiral image
>
> Target: f(x) is the vector (a.k.a. logits) input to softmax.
>
> AUROC is 0.508 when likelihood p(x) is used as OOD score
>
> AUROC is 0.502 when classification output max p(y|x) is used as OOD score
>
> AUROC is 0.804 when ||d_logp(x)/d_x|| is used as OOD score
>
> Note-1: the authors provided only the model pretrained on CIFAR10.
>
> Note-2: using random noise as the initial OOD sample, many generated images look like images in CIFAR10 dataset, and therefore, we used a color spiral image as the initial OOD sample.

---

> > ### Author Response · Authors · 2020-11-23
> > **update**
> >
> > a correction:   -||d_logp(x)/d_x|| is used in JEM as the OOD score, and the corresponding AUROC is 0.196. We were misled by Table 3 of the JEM paper, which shows ||d_logp(x)/d_x|| is the OOD score. To figure this out, we checked the source code of the method and the relevant descriptions in the paper.
> >
> > We redo the experiments using another set of parameters: ε=10, α= ε/100, N=1e3
> >
> > The initial OOD sample is a color spiral image
> >
> > Target: f(x) is the vector (a.k.a. logits) input to softmax.
> >
> > AUROC is 0.559 when likelihood p(x) is used as OOD score
> >
> > AUROC is 0.513 when classification output max p(y|x) is used as OOD score
> >
> > AUROC is 0.203 when - ||d_logp(x)/d_x|| is used as OOD score

---

> ### Author Response · Authors · 2020-11-17
> **New Experiment 8**
>
> [8] Glow NLL with input complexity (ICLR 2020, https://arxiv.org/abs/1909.11480)
>
> Dataset: CelebA subset that has 160 images
>
> DNN: Glow source code is from https://github.com/rosinality/glow-pytorch
>
> Parameters:  ε=10, α= ε/100, N=1e4
>
> The initial OOD sample is a color spiral image.
>
> Target: f(x) = NLL-L, where L is input complexity measured by PNG compression.
>
> AUROC is 0.500.
>
> Note: this method combines Glow negative log-likelihood (NLL) and input complexity. In our original manuscript, we show that Glow NLL score can be arbitrarily manipulated. In our new experiment, we show that NLL combined with input complexity can still be arbitrarily manipulated, which leads to the AUROC of 0.500.

---

> ### Author Response · Authors · 2020-11-19
> **New Experiment 9**
>
> [9] Gram (NeurIPS 2019, https://arxiv.org/abs/1912.12510)
>
> Datasets:  the first 500 images in CIFAR10 test set, the first 500 images in CIFAR100 test set.
>
> DNNs: pre-trained Resnet models with source code from https://github.com/VectorInstitute/gram-ood-detection
>
> Parameters:  ε=10, α= ε/100, N=100
>
> The initial OOD sample is a random noise image
>
> Target: f(x) = OOD_score (x) where OOD score is called Total Deviation.
>
> AUROC is 0.500 on CIFAR10 vs OOD; AUROC is 0.500 on CIFAR100 vs OOD; The results show that the OOD_score can be arbitrarily manipulated.
>
> Note: the method computes the p-th order gram matrix G=A^(1/p) with p in the range of 1 to p_max, which caused gradients to be inf or nan during back-propogation in the OOD attack algorithm. To resolve this problem, we tried three things:
>
> (a) use double precision (float64)
>
> (b) rewrite G = exp[(1/p)*log(A+eps)] where eps=1e-40
>
> (c) use (b) to generate images during OOD attack, and use  G=A^(1/p) to compute OOD_score.
>
> (a)&(b) work for p_max=5, for larger p_max, we still get numerical problem (inf or nan). From Figure 2 of the paper, the method has already achieved better performance compared to Mahalanobis when p_max is 5. Thus, p_max=5 is a valid choice for the evaluation of the method.

---

### Author Response · Authors · 2020-11-17
**Clarification**

We thank the reviewers for thorough reading and reviewing our manuscript.

Our contributions:

(1) We developed the OOD Attack algorithm to generate OOD samples starting from initial OOD samples, by using iterative projected gradient descent (PGD) optimization technique.

(2) We hypothesized that dimensionality reduction in an encoder provides the opportunity for the existence of the mapping of OOD and in-distribution samples to the same locations in the latent space. We applied the OOD Attack algorithm to evaluate DNN classifiers on various datasets, and the results confirmed the hypothesis.

(3) We evaluated nine OOD detection methods using the OOD Attack algorithm. The algorithm and the results can serve as a reference for the evaluation of new OOD detection methods.

Adversarial attacks vs OOD attacks:

Adversarial attacks and OOD attacks are doing completely different things to neural networks, although the attack algorithms may use similar optimization techniques. An adversarial attack will add a small amount of noise to the input, and the noisy image is still human-recognizable. For example, a noisy image of a panda is still an image of the same panda. By the judgment of humans, the noisy image and the clean image are the images of the same object, and they are in the same class. As a comparison, OOD samples, which can be generated by the OOD attack algorithm, can be arbitrary (e.g. random noise) as long as they do not look like in-distribution samples.

---

> ### Author Response · Authors · 2020-11-23
> **update**
>
> We deleted the part "a simple method for OOD detection with theoretical guarantee: subspace saturation training of bijective DNN for OOD detection" from the paper

---

> ### Author Response · Authors · 2020-11-23
> **update**
>
> The root causes of the existence of adversarial attacks and OOD attacks are quite different.
>
> Adversarial samples may exist if the decision boundaries are too close to the samples. Maximizing sample margins (i.e. moving decision boundaries far away from the samples during training) may eliminate the existence of many adversarial samples. (MMA, ICLR 2020 https://arxiv.org/abs/1812.02637)
>
> OOD attacks may succeed if the network (considering the operations in an OOD detection method as a part of the network) does dimensionality reduction, so that OOD and in-distribution samples can be mapped to the same locations in the latent space of the network. A bijective mapping alone (e.g. Glow) cannot defend against OOD attacks. If we have in-distribution and OOD samples "enough" for binary classification, a bijective mapping is not necessary.
>
> The results from New Experiment 5 (Appendix F in the revised paper) show that networks trained (using MMA, ICLR 2020 https://arxiv.org/abs/1812.02637) to be robust to adversarial attacks may not be robust to OOD attacks, which further highlights the difference between adversarial attacks and OOD attacks.
>
> To defend against OOD attacks, here are some possible strategies that we learned from this study:
>
> (1) reconstruction-based auto-encoder (Section 2.4 in the paper): OOD score is computed in the input space
> It has known limitations: it cannot guarantee that OOD samples will have large reconstruction errors.
>
> (2) no dimensionality reduction
>
>    (2.1) see the subspace saturation training method in the original manuscript (deleted in the revised paper)
>
>    (2.2) use GMMs to build density models in the input space as done by (Meinke & Hein, 2020), which has limitations
>
> (3) use in-distribution and OOD samples for implicit or explicit binary classification during training
>
>   see JEM in New Experiment 7 (Appendix J in the revised paper) for implicit binary classification

---

### Decision · Program_Chairs · 2021-01-07
**Final Decision**

**Decision:**

Reject

**Comment:**

**Problem significance** This paper proposes an attack mechanism in the latent space of a neural network f(x), which produces out-of-distribution examples. The AC agrees reviewers on the significance of the OOD detection problem, particularly addressing the vulnerability aspect is relevant and of great interest to the community.

**Technical contribution** The AC shares the concern with several reviewers on the limited technical novelty as well as the problem formulation. While the authors have clarified the difference between adversarial attack vs. OOD attack, the underlying attack mechanism is not new to the community (except for allowing for a larger degree of search space without constrained by the visual imperceptibility). In some sense, the search is made easier than the standard adversarial attack by removing the similarity constraint. Given the unrealisticness of the created OOD examples (largely noisy patches), the AC thinks perhaps a more interesting problem is to look at naturally occurring OOD examples that would lead to the similar latent encoding w.r.t in-distribution data, or adversarial robustness w.r.t the OOD detector.  This to me, would steer the community in the right direction.

From a problem formulation perspective, the AC thinks it's useful to differentiate three highly related attacks (that are distinct but can cause confusions):

- adversarial attack w.r.t the classifier
- OOD attack w.r.t the classifier
- adversarial attack w.r.t the OOD detector (see recent works [1][2][3] which considered the robustness aspect of OOD detector)


**Rebuttal feedback** The AC recognizes the effort made by the authors to address the concerns and comments raised by reviewers. The AC agrees with R1/R2/R3 that the additional experiments are valuable, however, the changes to the manuscript are substantial enough to deem another round of review in the future venue. The paper can improve with better organization and presentation, moving the results in the appendix to the main paper.

**Recommendation** The AC recommends rejection.

References

[1] Sehwag et al. Analyzing the robustness of open-world machine learning. 2019

[2] Hein et al. Why relu networks yield high-confidence predictions far away from the training data and how to mitigate the problem. 2019

[3] Chen et al. Informative Outlier Matters: Robustifying Out-of-distribution Detection Using Outlier Mining. arXiv:2006.15207